# Enhancing Trust in Large Language Models with Uncertainty-Aware Fine-Tuning

## Abstract

Large language models (LLMs) have revolutionized the field of natural language processing with their impressive reasoning and question-answering capabilities. However, these models are sometimes prone to generating credible-sounding but incorrect information, a phenomenon known as LLM hallucinations. Reliable uncertainty estimation in LLMs is essential for fostering trust in their generated responses and serves as a critical tool for the detection and prevention of erroneous or hallucinated outputs. To achieve reliable and well-calibrated uncertainty quantification in open-ended and free-form natural language generation, we propose an uncertainty-aware fine-tuning approach for LLMs. This approach enhances the model's ability to provide reliable uncertainty estimates without compromising accuracy, thereby guiding them to produce more trustworthy responses. We introduce a novel uncertainty-aware causal language modeling loss function, grounded in the principles of decision theory. Through rigorous evaluation on multiple free-form question-answering datasets and models, we demonstrate that our uncertainty-aware fine-tuning approach yields better calibrated uncertainty estimates in natural language generation tasks than fine-tuning with the standard causal language modeling loss. Furthermore, the experimental results show that the proposed method significantly improves the model's ability to detect hallucinations and identify out-of-domain prompts.

## 1 Introduction

Large Language Models (LLMs) have shown remarkable success in various natural language processing tasks (Touvron et al., 2023; Gemma et al., 2024; Achiam et al., 2023) and are increasingly becoming ubiquitous in a variety of domains for their decision-making and reasoning abilities (Eigner & Händler, 2024). However, their real-world deployment, particularly in high-stakes and safety-critical applications, is hindered by challenges such as hallucinations and out-of-domain prompts, which can lead to the generation of erroneous or nonsensical outputs. Hallucinations, often described as plausible-sounding but incorrect or unfaithful model generations (Ji et al., 2023), present a crucial challenge in developing trustworthy systems especially in critical domains such as medical (Ahmad et al., 2023) and legal (Magesh et al., 2024). The ability to recognize out-of-domain prompts and to acknowledge the limits of a model's knowledge base paves the way for building safe AI systems (Amodei et al., 2016).

Uncertainty quantification (UQ) in LLMs plays a pivotal role in understanding what the model knows and does not know, which is an active area of research for free-form natural language generation (NLG) (Kadavath et al., 2022; Kuhn et al., 2023; Lin et al., 2024). UQ methods has emerged as a step towards determining the trustworthiness of responses generated by LLMs (Fadeeva et al., 2023; Plaut et al., 2024; Kadavath et al., 2022). Uncertainty estimation techniques such as semantic entropy (Kuhn et al., 2023) have shown to be effective indicators in detecting 'confabulations' (Farquhar et al., 2024), a subcategory of hallucinations characterized by the generation of arbitrary and incorrect responses.

The calibration of uncertainty estimates is crucial for the reliability of LLMs; a well-calibrated model should correlate low uncertainty with accurate responses and high uncertainty with likely incorrect responses. However, recent studies (Xiong et al., 2024; Yang et al., 2024) have revealed that LLM predictions are often poorly calibrated, leading to overconfidence in incorrect outputs.

This problem is more pronounced in fine-tuned language models (Kong et al., 2020; Liu et al., 2024b). Unlike the pre-training phase, where models are exposed to vast amounts of unlabeled data, fine-tuning involves limited labeled data. Consequently, the immense capacity of LLMs can lead to overfitting on this limited data, producing overconfident predictions (Kong et al., 2020). This presents a substantial challenge, as the model may produce unreliable uncertainty metrics that are influenced by the model's miscalibrated token confidence. Moreover, evaluating calibration in NLG is particularly challenging due to the variable lengths of generated text compared to reference sentences. Traditional calibration error metrics (Naeini et al., 2015; Nixon et al., 2019) assume a fixed number of outcomes or classes, which aligns well with classification tasks but not with the open-ended nature of NLG, where the number of tokens in generated text and reference sentences can differ. To address this, we exploit the inverse correlation between uncertainty quantification and the quality of generated text, allowing us to perform calibration analysis in NLG settings.

In this work, we introduce an uncertainty-aware fine-tuning approach for LLMs, which is grounded in decision theory and tailored for free-form natural language generation. Our approach is orthogonal to existing uncertainty quantification methods and is driven by the goal of enhancing the reliability of uncertainty metrics through uncertainty-aware fine-tuning. Specifically, we achieve this with an optimization objective that encourages the model to learn to associate high uncertainty with incorrectly generated tokens and low uncertainty with correctly generated tokens, while maximizing accuracy within the framework of causal language modeling. We show that fine-tuning with our calibration objective enhances the reliability of uncertainty quantification in LLMs. The calibrated uncertainty estimates serve as a crucial tool for enhancing the trustworthiness of generated responses, represents a substantial step towards identifying hallucinations, and improving decision-making capabilities through selective generation (Ren et al., 2022).

Our contributions are as follows:

- We propose uncertainty-aware causal language modeling (UA-CLM) loss function, designed for fine-tuning LLMs to produce well-calibrated uncertainty estimates in free-form natural language generation.

- We conduct uncertainty calibration analysis in free-form NLG settings, employing an innovative methodology that leverages the inverse correlation between uncertainty quantification and the quality of the generated text.

- We perform a comprehensive empirical evaluation across four key aspects: hallucination detection; selective generation; out-of-prompt detection; and calibration analysis, demonstrating that UA-CLM significantly enhances the quality of uncertainty estimates in LLMs in open-ended and free-form question-answering tasks. Notably, these enhancements in uncertainty calibration are achieved without compromising the accuracy when compared to standard CLM.

- In addition, we have also applied our proposed UA-CLM methodology to a large vision language model (LVLM), demonstrating its efficacy in the open-ended visual question-answering task. This extension of our work shows the versatility of UA-CLM in handling multimodal inputs and complex tasks beyond text-based question-answering, showcasing it's utilty to wider range of applications.

## 2 BACKGROUND AND RELATED WORKS

### 2.1 UNCERTAINTY ESTIMATION IN NATURAL LANGUAGE GENERATION

We refer to Abdar et al. (2021); Gawlikowski et al. (2023) for surveys on uncertainty quantification (UQ) in deep learning. In machine learning models, predictive uncertainty is composed of two primary sources: epistemic uncertainty, associated with model's lack of knowledge and aleatoric uncertainty - the inherent noise in the data or observation. As LLMs continue to evolve rapidly, there is an increasing interest in enhancing our understanding of the uncertainty associated with LLM responses for developing trustworthy and reliable systems (Fadeeva et al., 2023). UQ methods from deep learning can be effectively applied to structured natural language processing tasks, such as text classification (Xiao & Wang, 2019) and multiple-choice question answering (Kumar et al., 2023). However, the application of these methods to free-form natural language generation presents distinct challenges.

The landscape of UQ in free-form NLG is a dynamic field of research, where methodologies are generally bifurcated into white-box (Fomicheva et al., 2020; Kuhn et al., 2023) and black-box approaches (Lin et al., 2024; Xiong et al., 2024). White-box methods necessitate access to the model's logits, or likelihood scores, or other internals of LLMs. Black-box methods rely solely on the analysis of the text sequences generated by the LLMs. Recent work (Tian et al., 2023) explores prompting techniques in reinforcement learning from human feedback (RLHF) language models to explicitly elicit verbalized response confidence. In contrast, another line of research has explored unsupervised methods for UQ from the models by utilizing confidence (Plaut et al., 2024), perplexity (Fomicheva et al., 2020), token entropy (Malinin & Gales, 2021) and semantic entropy (Kuhn et al., 2023) to quantify uncertainty in LLM responses. Previous research (Minderer et al., 2021) has shown that confidence or entropy measures can be susceptible to poor calibration and may not fully reflect a model's underlying uncertainties. Our work focuses on improving these uncertainty metrics in white-box settings by better calibrating the language models with uncertainty-aware finetuning.

## 2.2 MODEL CALIBRATION

Calibration is important in applications where decision-making relies on not just the model's predictions, but also on the trustworthiness of its uncertainty scores. It is important to capture well-calibrated uncertainty estimates of a model for creating reliable and trustworthy systems. Model calibration is a well-explored area of research in deep learning, with a variety of strategies proposed to enhance the calibration of deep neural networks for classification and regression tasks. These strategies include post-hoc rescaling techniques (Guo et al., 2017; Kull et al., 2017), which adjust the model's predictions to better align with true event likelihoods; data augmentation, which enriches the training dataset to promote generalization (Thulasidasan et al., 2019; Hendrycks et al., 2020); and probabilistic modeling approaches that integrate uncertainty directly into the model's architecture (Blundell et al., 2015; Lakshminarayanan et al., 2017). Other line of works utilize explicit calibration loss functions during training to directly optimize for calibration (Kumar et al., 2018; Krishnan & Tickoo, 2020; Mukhoti et al., 2020; Karandikar et al., 2021), that has resulted in better calibrated models.

Calibration of LLMs for natural language processing tasks is an ongoing area of research. Prior works have largely focused on refining LLMs for structured tasks like text classification (Kong et al., 2020) or multiple-choice question answering (Desai & Durrett, 2020; Jiang et al., 2021). Studies like the one by Xiong et al. (2024) have highlighted that despite the impressive performance of foundational LLMs on a wide array of tasks, these models often exhibit poor calibration, particularly exhibiting overconfidence in their predictions. As LLMs are increasingly used in natural language generation tasks, new calibration techniques (Geng et al., 2024) are emerging to enhance the reliability of the generated text. More recently, Liu et al. (2024b) introduced a calibration technique for LLMs that trains a single linear layer over the model's last hidden layer representations to predict a bias term, which is then used to adjust the model's logits and alter the generation confidence for short-form and long-form responses. Band et al. (2024) propose a training objective for linguistic calibration, utilizing reinforcement learning to optimize and calibrate long-form text generations. Kapoor et al. (2024) proposed a calibration tuning method designed for LLMs in multiple-choice question-answering settings. Prior work by Liu et al. (2024b) has shown that standard fine-tuning of LLMs can lead to poorer calibration. The calibration of LLMs for free-form text generation, as well as uncertainty-aware fine-tuning, represents a significant open area of research. Our work addresses this gap by developing an uncertainty-aware fine-tuning method of LLMs, fine-tuning less than 1% of the model parameters, to achieve well-calibrated models for free-form natural language generation.

## 2.3 FINE-TUNING LARGE LANGUAGE MODELS

With the emergence of foundation models, fine-tuning have become a common practice in the field of natural language processing, enabling the adaptation of general-purpose pre-trained models to specialized tasks and domains. As fine-tuning a LLM with billions of parameters can be resource-intensive, parameter-efficient fine-tuning (Mangrulkar et al., 2022) strategies have been proposed. These parameter-efficient fine-tuning techniques also mitigate catastrophic forgetting more effectively in comparison to full fine-tuning (Wang et al., 2022). One approach is to update only a subset of the model's parameters, such as adapter modules (Houlsby et al., 2019) or Low-Rank Adaptation

(LoRA) (Hu et al., 2022), where only a small set of parameters are updated while the pre-trained weights are frozen. Another strategy is prompt-based fine-tuning (Liu et al., 2023), where models are conditioned on task-specific prompts to guide the text generation without model parameter updates. We leverage LoRA strategy to illustrate our proposed uncertainty-aware finetuning in this work.

## 3 UNCERTAINTY-AWARE CAUSAL LANGUAGE MODELING

Motivated by the need to overcome the challenges of uncertainty miscalibration (Xiong et al., 2024) in Large Language Models (LLMs) and the increasing trend of fine-tuning pre-trained foundational models for domain-specific adaptation — where fine-tuned LLMs often exhibit overconfidence in their predictions (Kong et al., 2020) — we propose a novel uncertainty calibration fine-tuning approach for natural language generation settings. We introduce a novel uncertainty-aware causal language modeling loss based on the principles of decision theory (Murphy, 2012). Our fine-tuning approach emphasizes increasing the uncertainty for wrong token predictions, while optimizing for accuracy and certainty for correct token predictions. Decision theory offers a mathematical and theoretical framework that guides to achieve optimal predictions by employing a task-specific utility function. Within the decision theory framework, our task is to generate natural language text accompanied by reliable uncertainty estimates. The utility function in this scenario is represented by the uncertainty-aware optimization objective function that is aimed at producing well-calibrated uncertainty estimates for causal language modeling. We design a differentiable loss function that incentivizes the model to yield low uncertainty when it generates correct tokens, and encourages the model to exhibit high uncertainty when it is at risk of predicting the next token incorrectly.

In causal language modeling, the goal is to predict the next token in a sequence given the previous tokens. Given a sequence of tokens $[w_1, w_2, \ldots, w_T]$, where T is the length of the sequence and each token $w_i$ is an element from a fixed vocabulary of size V, the model aims to learn the conditional probability distribution $P_\theta(w_i|w_{0:i-1})$ for each token $w_i$ given the preceding set of tokens $w_{0:i-1}$; where, $\theta$ represents the parameters of the LLM. The loss function for standard causal language modeling (CLM) is typically the negative log-likelihood as defined in Equation 1 below.

$$\mathcal{L}_{\text{CLM}}^\theta := -\frac{1}{T} \sum_{i=0}^{T} \log P_\theta(w_i|w_{0:i-1}) \tag{1}$$

**Desideratum:** The desired and ideal outcome in causal language modeling is to achieve a state where every correctly generated token is assigned low predictive uncertainty, and high predictive probability, reflecting the model's high confidence in its accuracy. Conversely, for every token that is generated incorrectly, the model should assign high uncertainty, and low predictive probability, denoting low confidence in these instances. This ensures that the model's confidence levels and uncertainty estimates are perfectly calibrated with the actual correctness of its predictions.

We define the uncertainty-aware causal language modeling (UA-CLM) loss in Equation 2 based on the above desideratum. The loss function captures the trade-off between predictive accuracy and uncertainty calibration, and is composed of two terms: one that deals with incorrectly generated tokens and one that deals with correctly generated tokens.

$$\mathcal{L}_{\text{UA-CLM}}^\theta := \underbrace{-\frac{1}{|\widetilde{C}|} \sum_{i \in \widetilde{C}} P_\theta(w_i|w_{0:i-1}) \log\left(\tanh\left(H_i\right)\right)}_{\text{Utility function for incorrect tokens}}$$
$$\underbrace{-\frac{1}{|C|} \sum_{i \in C} \left(1 - P_\theta(w_i|w_{0:i-1})\right) \log\left(1 - \tanh\left(H_i\right)\right)}_{\text{Utility function for correct tokens}} \tag{2}$$

where,

$$H_i := -\sum_{j=1}^{V} P_\theta(w_i^j|w_{0:i-1}) \log P_\theta(w_i^j|w_{0:i-1}) \tag{3}$$

Here, $H_i$ is the entropy of the probability distribution of the $i^{th}$ token $w_i$ given the previous tokens $w_{0:i-1}$ in the sequence, $\overline{w_i}$ is the ground-truth reference token, $C := \{i \mid w_i = \overline{w_i}\}$ is a set of indices corresponding to correctly predicted tokens, $\widetilde{C} := \{i \mid w_i \neq \overline{w_i}\}$ is a set of indices corresponding to incorrectly predicted tokens, V is the size of the vocabulary, $w_i^j$ is the $j^{th}$ token in the vocabulary, and $P_\theta(w_i^j|w_{0:i-1})$ is the predicted probability of the $j^{th}$ token in the vocabulary. The hyperbolic tangent function is employed to scale the token entropy values due to its smooth gradient properties (Krishnan & Tickoo, 2020), ensuring that $\tanh(H_i)$ lies in the interval [0, 1].

The loss function in Equation 2 offers theoretical guarantees as an optimization objective and satisfies the desideratum, converging to a perfect value of zero when all correctly generated tokens have a predictive probability of 1 (indicating high confidence) and scaled predictive entropy of 0 (implying low uncertainty), while all incorrectly generated tokens have a predictive probability of 0 (representing low confidence) and scaled predictive entropy of 1 (implying high uncertainty). The differentiable utility functions in the UA-CLM loss as shown in Equation 2 steers the predictive probabilities and uncertainty estimates to align with the accuracy of subsequent token predictions in autoregressive models. When the uncertainty estimates in the predicted tokens are misaligned, the loss increases, thereby directing the stochastic gradient computations to drive the loss towards minimization. This loss reduces when the uncertainty estimates conform to the desideratum, enabling the model to produce well-calibrated uncertainties while maximizing accuracy.

---

**Algorithm 1** Uncertainty-aware fine-tuning in LLMs

---

1: **Input:** Pre-trained LLM $\mathbb{M}$ with parameters $\phi$, LoRA parameters $\theta$, learning rate $\eta$, epochs $E$, training data $\mathcal{D}$
2: **Output:** Uncertainty calibrated fine-tuned LLM $\mathbb{M}'$
3: Initialize LoRA parameters $\theta$, and freeze $\phi$
4: **for** $epoch = 1$ to $E$ **do**
5:      **for** each batch $B \subseteq \mathcal{D}$ **do**
6:          Compute forward pass to get $P_{\phi+\theta}(w_i|w_{0:i-1})$ and token uncertainty estimate $H_i$
7:          Compute UA-CLM loss $\mathcal{L}_{\text{UA-CLM}}^{\phi+\theta}$               ▷ Equation 2
8:          Compute gradients of loss function w.r.t. $\theta$, $\nabla_\theta \mathcal{L}_{\text{UA-CLM}}^{\phi+\theta}$
9:          Update LoRA parameters: $\theta \leftarrow \theta - \eta \cdot \nabla_\theta \mathcal{L}_{\text{UA-CLM}}^{\phi+\theta}$
10:      **end for**
11: **end for**
12: $\theta^* \leftarrow \theta$
13: $\mathbb{M}' \leftarrow \mathbb{M}$ with updated LoRA parameters $\theta^*$
14: **return** $\mathbb{M}'$

---

Our proposed UA-CLM loss is designed to be agnostic to various parameter-efficient fine-tuning methods in LLMs; however, in this paper, we leverage and illustrate its application through Low-Rank Adaptation (LoRA) (Hu et al., 2022) as described in Algorithm 1. By incorporating uncertainty directly into the loss function, the model can not only learn to improve accuracy but also to understand a meaningful representation of uncertainty in its predictions. This dual emphasis on accuracy and uncertainty in the optimization objective ensures that the model's uncertainty estimates are closely aligned with the actual predictive accuracy of the generated tokens, leading to improved uncertainty calibration in natural language generation.

## 4 EXPERIMENTS AND RESULTS

We perform extensive empirical evaluation to compare our proposed uncertainty-aware causal language modeling (UA-CLM) fine-tuning method to the standard causal language modeling (CLM) fine-tuning, pre-trained baseline, unlikelihood training (ULT) (Welleck et al., 2020), and calibration tuning (CT) (Kapoor et al., 2024) methods. We evaluate on open-ended, and free-form natural language generation tasks. Our comprehensive evaluation rigorously assesses the quality of uncertainty estimates and the quality of the generated text. This includes an analysis of broadly four aspects: hallucination detection, uncertainty-guided selective generation, out-of-domain prompt detection, and calibration analysis based on the inverse correlation between the uncertainty estimates and the quality of generated text.

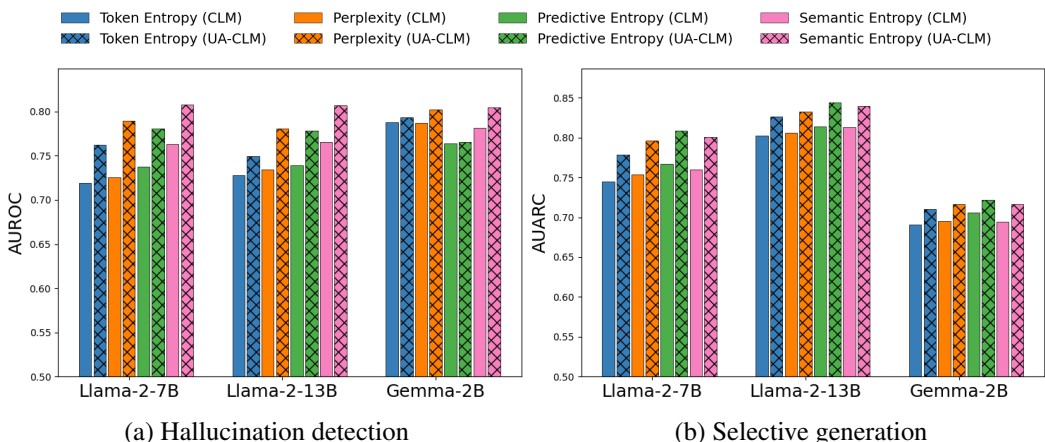

(a) Hallucination detection          (b) Selective generation

Figure 1: The proposed Uncertainty-aware Causal Language Modeling (UA-CLM) outperforms standard Causal Language Modeling (CLM) in all four UQ metrics across various models. The performance is evaluated using AUROC for hallucination detection and AUARC for selective generation based on four distinct UQ metrics.

### 4.1 EXPERIMENTAL SETTINGS

**Datasets** We utilize free-form question-answering (QA) datasets to evaluate the proposed methods on LLMs: CoQA (Reddy et al., 2019), an open-book conversational QA dataset; TriviaQA (Joshi et al., 2017), a reading comprehension QA dataset. These datasets are frequently utilized benchmarks for evaluating uncertainty quantification (UQ) in LLMs for natural language generation, as evidenced by prior works from Kuhn et al. (2023), Lin et al. (2024), and Farquhar et al. (2024). We employ the OK-VQA dataset (Marino et al., 2019), an open-ended visual question-answering (VQA) dataset to extend our evaluation to large vision language models (LVLMs), thereby providing a comprehensive analysis of our approach across diverse open-ended free-form QA tasks. Additionally, we use BioASQ (Krithara et al., 2023), a biomedical question-answering dataset for the evaluation of out-of-domain prompt detection. In our experiments, we utilize the development split of the CoQA dataset, which contains approximately 8,000 question-answer pairs, the validation split of TriviaQA with around 10,000 question-answer pairs, and the validation split of OK-VQA, comprising roughly 5,000 question-answer pairs along with their corresponding images. For each dataset, we allocate 20% of the data for fine-tuning purposes, while the remaining 80% serve as the test sets for evaluation. We use a standard prompt across all datasets, more details on the datasets and the prompt are provided in Appendix A.1.1 and A.1.2, respectively.

**Models** We use the Llama-2 models with 7B and 13B parameters (Touvron et al., 2023) and the Gemma model with 2B parameters (Gemma et al., 2024) for the free-form QA experiments. Additionally, we utilize the LLaVA-1.5 model with 7B parameters (Liu et al., 2024a) for the open-ended visual question-answering task.

**Fine-tuning** We perform parameter-efficient fine-tuning using the Low-rank Adaptation (LoRA) framework (Hu et al., 2022). The models undergo fine-tuning as described in Section 3 for the uncertainty-aware causal language modeling method. For comparative purposes, we also fine-tune models using the standard cross-entropy loss to evaluate against the standard causal language modeling fine-tuning method. For all our experiments, the models are fine-tuned for a concise duration of 3 epochs, utilizing only 20% of the data split. The optimization is carried out using the AdamW optimizer (Loshchilov & Hutter, 2019), with an initial learning rate of $1e\text{-}4$, a weight decay of $0.001$, and a warm-up ratio of $0.03$. We follow the same setup for both CLM and UA-CLM methods for a fair comparison. During the fine-tuning process, only the LoRA parameters are updated, while all other model parameters remain frozen. We provide more details on the hyperparameters and implementation in Appendix A.1.3, to facilitate the reproducibility of the results.

Table 1: Evaluation of uncertainty quantification: Comparative analysis of the proposed Uncertainty-aware Causal Language Modeling (UA-CLM) with standard Causal Language Modeling (CLM), pre-trained baseline, UnLikelihood Training (ULT) (Welleck et al., 2020), and Calibration Tuning (CT) (Kapoor et al., 2024) methods (the best values are in **bold**). The comparison spans different datasets and models, with quality of uncertainty quantification evaluated using the Area Under the Receiver Operating Characteristic (AUROC) and the Area Under the Accuracy-Rejection Curve (AUARC) based on four different uncertainty metrics.

| Dataset | Model | Finetuning Method | AUROC ↑ (Hallucination detection) | | | | AUARC ↑ (Area under accuracy-rejection curve) | | | |
|---|---|---|---|---|---|---|---|---|---|---|
| | | | Token Entropy | Perplexity | Predictive Entropy | Semantic Entropy | Token Entropy | Perplexity | Predictive Entropy | Semantic Entropy |
| CoQA | Llama-2-7B | Pre-trained | 0.5813 | 0.6324 | 0.6686 | 0.7467 | 0.8361 | 0.8606 | 0.9348 | 0.9411 |
| | | CLM | 0.6252 | 0.6320 | 0.6635 | 0.6889 | 0.9435 | 0.9444 | 0.9508 | 0.9530 |
| | | ULT | 0.5790 | 0.5915 | 0.6793 | 0.6495 | 0.9212 | 0.9219 | 0.9315 | 0.9326 |
| | | CT | 0.6175 | 0.6571 | 0.6706 | 0.7292 | 0.8603 | 0.8780 | 0.9029 | 0.9075 |
| | | UA-CLM | **0.6955** | **0.7398** | **0.7413** | **0.7741** | **0.9603** | **0.9657** | **0.9699** | **0.9716** |
| | Llama-2-13B | Pre-trained | 0.6027 | 0.6404 | 0.6679 | 0.7111 | 0.8672 | 0.8940 | 0.9137 | 0.9209 |
| | | CLM | 0.6302 | 0.6348 | 0.6815 | 0.6910 | 0.9579 | 0.9584 | 0.9659 | 0.9661 |
| | | ULT | 0.6323 | 0.6523 | 0.6883 | 0.7158 | 0.9510 | 0.9534 | 0.9592 | 0.9612 |
| | | CT | 0.5299 | 0.5599 | 0.6072 | 0.6958 | 0.8497 | 0.8647 | 0.8944 | 0.9200 |
| | | UA-CLM | **0.6701** | **0.7255** | **0.7363** | **0.7694** | **0.9645** | **0.9700** | **0.9784** | **0.9792** |
| | Gemma-2B | Pre-trained | 0.7073 | 0.7089 | 0.6962 | 0.7635 | 0.9271 | 0.9339 | 0.9235 | 0.9452 |
| | | CLM | 0.7723 | 0.7606 | 0.7295 | 0.7618 | 0.9468 | 0.9454 | 0.9619 | 0.9655 |
| | | ULT | 0.7097 | 0.6921 | 0.6540 | 0.7162 | 0.9172 | 0.9152 | 0.9093 | 0.9168 |
| | | UA-CLM | **0.7780** | **0.7837** | **0.7358** | **0.7871** | **0.9652** | **0.9668** | **0.9671** | **0.9672** |
| TriviaQA | Llama-2-7B | Pre-trained | 0.7687 | 0.8220 | 0.8191 | 0.8315 | 0.8050 | 0.8259 | 0.8251 | 0.8369 |
| | | CLM | 0.8135 | 0.8192 | 0.8108 | 0.8371 | 0.8617 | 0.8615 | 0.8558 | 0.8630 |
| | | ULT | 0.7676 | 0.8003 | 0.8004 | 0.8276 | 0.8273 | 0.8418 | 0.8448 | 0.8519 |
| | | CT | 0.7714 | 0.8211 | 0.8037 | 0.8233 | 0.8571 | 0.8812 | 0.8769 | 0.8834 |
| | | UA-CLM | **0.8293** | **0.8393** | **0.8197** | **0.8423** | **0.8879** | **0.8927** | **0.8780** | **0.8934** |
| | Llama-2-13B | Pre-trained | 0.7984 | 0.8123 | 0.7801 | 0.8365 | 0.8441 | 0.8574 | 0.8584 | 0.8587 |
| | | CLM | 0.8264 | 0.8333 | 0.7971 | 0.8407 | 0.8798 | 0.8807 | 0.8708 | 0.8829 |
| | | ULT | 0.8240 | **0.8485** | **0.8245** | **0.8456** | 0.8949 | 0.9055 | 0.9013 | 0.9063 |
| | | CT | 0.7338 | 0.7897 | 0.7991 | 0.8222 | 0.8460 | 0.8914 | 0.9007 | 0.9019 |
| | | UA-CLM | **0.8297** | 0.8352 | 0.8033 | 0.8447 | **0.9200** | **0.9254** | **0.9155** | **0.9252** |
| | Gemma-2B | Pre-trained | 0.7633 | 0.7719 | 0.7920 | 0.8127 | 0.6912 | 0.7225 | 0.7127 | 0.7279 |
| | | CLM | 0.8030 | 0.8138 | **0.7989** | 0.8018 | 0.7256 | 0.7251 | 0.7162 | 0.7198 |
| | | ULT | 0.7935 | 0.8134 | 0.7912 | 0.8035 | 0.7212 | 0.7429 | 0.7246 | 0.7413 |
| | | UA-CLM | **0.8085** | **0.8211** | 0.7960 | **0.8228** | **0.7373** | **0.7436** | **0.7258** | **0.7453** |
| OK-VQA | LLaVA-1.5-7B | CLM | 0.5504 | 0.5419 | 0.5455 | 0.5370 | 0.5809 | 0.5781 | 0.5790 | 0.5747 |
| | | UA-CLM | **0.6001** | **0.5984** | **0.6106** | **0.6638** | **0.5989** | **0.5965** | **0.6012** | **0.6265** |

Table 2: Generated text quality and calibration evaluation: Comparative analysis of Uncertainty-aware Causal Language Modeling (UA-CLM) fine-tuning method with standard Causal Language Modeling (CLM) fine-tuning, pre-trained baseline, UnLikelihood training (ULT) (Welleck et al., 2020) and Calibration Tuning (CT) (Kapoor et al., 2024) methods. The results in the table indicate that UA-CLM achieves higher ROUGE-L and accuracy, and lower expected calibration error (ECE) as compared to other methods.

| Finetuning Method | Llama-2-7B (CoQA) | | | Llama-2-7B (TriviaQA) | | | Llama-2-13B (CoQA) | | | Llama-2-13B (TriviaQA) | | |
|---|---|---|---|---|---|---|---|---|---|---|---|---|
| | ROUGE-L ↑ | Accuracy ↑ | ECE ↓ | ROUGE-L ↑ | Accuracy ↑ | ECE ↓ | ROUGE-L ↑ | Accuracy ↑ | ECE ↓ | ROUGE-L ↑ | Accuracy ↑ | ECE ↓ |
| Pret-trained | 0.7449 | 0.8350 | 0.0561 | 0.6654 | 0.7048 | 0.2304 | 0.7832 | 0.855 | 0.0559 | 0.7160 | 0.7610 | 0.2133 |
| CLM | **0.8886** | 0.9253 | 0.0343 | 0.6037 | 0.6529 | 0.2407 | 0.9106 | 0.9406 | 0.0323 | 0.6588 | 0.6967 | 0.2241 |
| ULT | 0.8409 | 0.8950 | 0.0588 | 0.6121 | 0.6586 | 0.3111 | 0.8771 | 0.925 | 0.0595 | 0.6875 | 0.7309 | 0.1517 |
| CT | 0.7437 | 0.8125 | 0.0410 | 0.6600 | 0.6987 | 0.2276 | 0.8022 | 0.8725 | 0.0992 | 0.7018 | 0.7429 | 0.1937 |
| UA-CLM | **0.8882** | **0.9264** | **0.0094** | **0.6679** | **0.7108** | **0.2090** | **0.9118** | **0.9461** | **0.0084** | **0.7277** | **0.7710** | **0.1365** |

**UQ metrics**   To assess the uncertainty quantification from CLM and UA-CLM in the context of free-form text generation, we employ four widely used metrics as the baselines: mean token entropy (Fomicheva et al., 2020), perplexity (Fadeeva et al., 2023), predictive entropy (Malinin & Gales, 2021) and semantic entropy (Kuhn et al., 2023). The predictive entropy and semantic entropy are estimated by generating 5 stochastic sequences from the model, each obtained through temperature sampling with a temperature setting of T=0.3.

## 4.2 EVALUATION AND RESULTS

**Hallucination detection**   We evaluate the performance of detecting confabulations (hallucinations) in the generated text using uncertainty estimates. Confabulations (Farquhar et al., 2024; Berrios, 1998) are a subset of hallucinations, characterized by LLMs making fluent claims that are

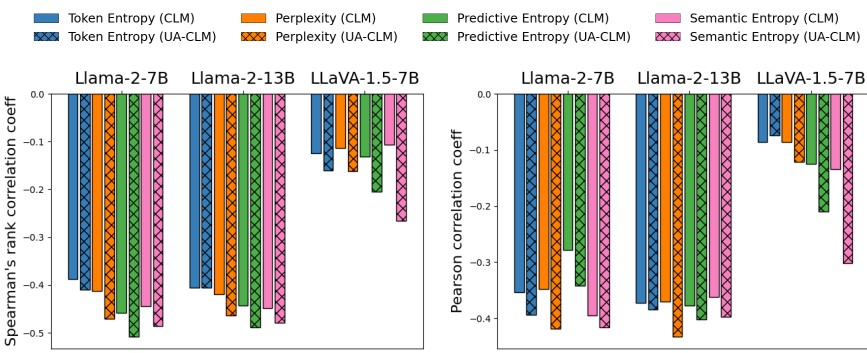

Figure 2: Uncertainty calibration analysis: Spearman's rank correlation coefficient and Pearson correlation coefficient between uncertainty estimates and generated text quality (ROUGE-L) scores for free-form open-ended question answering. Stronger negative correlation is desired for well-calibrated uncertainty quantification.

both wrong and arbitrary. Hallucination detection is a binary classification task to distinguish between correct and hallucinated (incorrect) responses based on the uncertainty estimate. We use Area Under the Receiver Operating Characteristic (AUROC) (Davis & Goadrich, 2006) to evaluate the quality of uncertainty quantification in terms of the model's capability to detect hallucinations, following the methodology used in (Farquhar et al., 2024). A higher AUROC indicates that the model is more effective at identifying correct responses and flagging hallucinations. The bar plot depicted in Figure 1(a) provides a visual representation of the AUROC performance for both the CLM and UA-CLM methods. It shows the improvement in various uncertainty quantification metrics across three distinct LLMs, underscoring the enhanced reliability of uncertainty estimates in the generated text achieved by UA-CLM. This plot consolidates the AUROC performance metrics from the CoQA and TriviaQA datasets, with specific numbers for each dataset and each LLM provided in Table 1. This table also includes the results for the OK-VQA dataset when evaluated with LVLM, offering a comprehensive study across different datasets, models, and uncertainty quantification metrics. We observe that the UA-CLM method exhibits a significant improvement in hallucination detection performance of up to 17.1% on QA tasks, and upto 23.6% on VQA task, over the standard CLM method.

Notably, these enhancements with UA-CLM are achieved without compromising quality of the generated text or the overall accuracy as presented in Table 2. We refer to Appendix A.2 for details of text quality metrics, and additional evaluation metrics can be found in Appendix A.4.

**Uncertainty-guided selective generation** The ability of a large language model to decide when to generate a response and when to abstain from providing one, based on its uncertainty estimates is crucial for building trustworthy and reliable generative AI models. This capability enables models to recognize and communicate their limitations. Selective generation (Ren et al., 2022) plays an important role in scenarios where providing an incorrect response could have negative consequences, such as in medical diagnosis, legal advice, or safety-critical information systems. We adopt the methodology proposed by Farquhar et al. (2024) and utilize the Area Under the Accuracy-Rejection Curve (AUARC) (Nadeem et al., 2009) to evaluate both the performance of the model and the quality of uncertainty estimates in the context of selective generation. AUARC serves as a valuable metric for evaluating the quality of a model's uncertainty estimates and its decision-making ability regarding when to make predictions and when to abstain due to high uncertainty. The bar plots presented in Figure 1(b), along with the numerical data provided in Table 1 shows a significant improvement in the AUARC scores achieved by the UA-CLM. This notable enhancement in AUARC indicates the uncertainty estimates with UA-CLM's can lead to better informed downstream decision-making.

**Correlation between uncertainty estimates and generated text quality** Calibration serves as a mechanism for ensuring the quality of uncertainty estimates. It is not feasible to assess the generated tokens calibration due to the potential mismatch in the number of tokens between the gen-

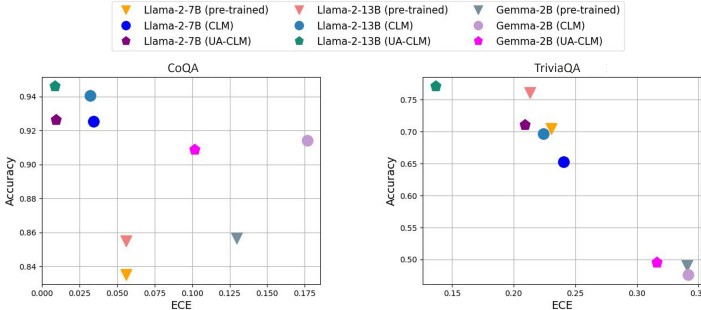

Figure 3: Accuracy versus Expected Calibration Error (ECE) comparison between UA-CLM, CLM, and pre-trained baseline across different LLM architectures on CoQA and TriviaQA datasets. The ideal model should have high accuracy and low ECE, indicating accurate predictions with well-calibrated uncertainty quantification (upper-left of the plot). The ECE of models fine-tuned with UA-CLM shows significant improvement compared to the pre-trained baseline and CLM fine-tuning.

Table 3: Out-of-domain detection: Evaluation with Biomedical question answering (BioASQ) as out-of-domain dataset on Llama-2-7B finetuned with CoQA dataset. The table shows the comparison of CLM and UA-CLM with AUROC and AUPR scores for out-of-domain detection using different uncertainty metrics.

| Method | AUROC ↑ (Out-of-domain detection) | | | | AUPR ↑ (Out-of-domain detection) | | | |
|---|---|---|---|---|---|---|---|---|
| | Token Entropy | Perplexity | Predictive Entropy | Semantic Entropy | Token Entropy | Perplexity | Predictive Entropy | Semantic Entropy |
| CLM | 0.7828 | 0.7661 | 0.7500 | 0.7902 | 0.8124 | 0.7928 | 0.8570 | 0.8422 |
| UA-CLM | **0.9025** | **0.8931** | **0.7763** | **0.9061** | **0.9235** | **0.9117** | **0.8958** | **0.9250** |

erated text and the ground truth reference. Hence we employ Spearman's rank correlation coefficient (Zwillinger & Kokoska, 1999) and Pearson correlation coefficient (Benesty et al., 2009) to evaluate the reliability of uncertainty estimates and examine how well these estimates align with the quality of the generated text, this is a novel methodology for calibration analysis to circumvent the challenges posed by varying token lengths in the generative outputs. We analyze the negative correlation between uncertainty estimates and ROUGE-L (Lin & Och, 2004) scores, a widely recognized metric for gauging generated text quality. A model with well-calibrated uncertainty estimates should demonstrate a strong negative correlation between uncertainty estimates and the generated text quality. As depicted in Figure 2, our findings from the CoQA, TriviaQA and OK-VQA datasets reveal a consistent negative correlation: the uncertainty estimates increase as the ROUGE-L scores tend to decrease, and vice-versa. The results indicate that fine-tuning with UA-CLM show a more pronounced inverse correlation between uncertainty and text quality compared to standard fine-tuning methods. Further, we estimate the sentence-level calibration with Expected Calibration Error (ECE) (Naeini et al., 2015) based on the correctness of generated response. The results in Table 2 shows that UA-CLM fine-tuning yields lower ECE as compared to other methods. Figure 3 shows the accuracy versus ECE plots for CoQA and TriviaQA datasets across different models for pre-trained baseline, CLM and UA-CLM fine-tuning methods. These results from the calibration analysis demonstrates the effectiveness of uncertainty-aware fine-tuning to obtain better calibrated uncertainty in free-form text generation tasks.

Table 4: Evaluating generalizability of fine-tuning methods on QA task and biography generation task.

| Method | CoQA −>TriviaQA | | | | TriviaQA −>CoQA | | | | CoQA −>BioGen | | | | |
|---|---|---|---|---|---|---|---|---|---|---|---|---|---|
| | AUROC ↑ (Hallucination detection) | | | | AUROC ↑ (Hallucination detection) | | | | BERT F1 ↑ | ECE ↓ | AUROC ↑ (Hallucination detection) | | |
| | Token Entropy | Perplexity | Predictive Entropy | Semantic Entropy | Token Entropy | Perplexity | Predictive Entropy | Semantic Entropy | | | Token Entropy | Perplexity | Semantic Entropy |
| CLM | 0.7201 | 0.7847 | 0.7532 | 0.7874 | 0.5952 | 0.6349 | 0.6665 | 0.7146 | 0.7394 | 0.2511 | 0.5653 | 0.5793 | 0.5281 |
| UA-CLM | **0.8271** | **0.8261** | **0.7880** | **0.8146** | **0.6456** | **0.6824** | **0.7154** | **0.7528** | **0.7405** | **0.1713** | **0.6135** | **0.6123** | **0.5354** |

**Out-of-domain detection**    In our experiments, we assess the model's capability to identify whether a given prompt is out-of-domain, referring to a question or input that falls outside the scope of model's trained knowledge base. To quantify this ability, we employ two widely recognized metrics: the Area Under the Receiver Operating Characteristic curve (AUROC) and the Area Under the Precision-Recall curve (AUPR) (Saito & Rehmsmeier, 2015). The Biomedical question-answering (BioASQ) (Krithara et al., 2023) dataset serves as our out-of-domain dataset, while the Conversational Question Answering (CoQA) dataset is used to represent in-domain data. We leverage uncertainty metrics as a means to detect out-of-domain prompts effectively. The results, as detailed in Table 3, demonstrate that our UA-CLM significantly outperforms the standard CLM in out-of-domain detection tasks. This performance is consistently observed across all four uncertainty metrics employed in the study, with up to 16.5% improvement in AUROC and up to 15% improvement in AUPR scores.

**Generalization and long-form text generation**    We conducted experiments to evaluate the generalizability of the proposed uncertainty-aware CLM fine-tuning method. Since we fine-tune the model in a causal language modeling setup that involves next-token prediction, the learning should be transferable, thereby encouraging generalizability. We evaluated the model fine-tuned with CoQA dataset on TriviaQA dataset, and vice-versa. Additionally, to evaluate the generalization beyond QA tasks, we conducted experiments for biography generation, a long-form paragraph-level generation task, following the recent works (Liu et al., 2024b; Band et al., 2024). The Llama-2-7B models fine-tuned with CLM and UA-CLM on CoQA were given prompts to write biographies of popular figures, whose names are sourced from BioGen (Min et al., 2023). The generated responses were compared against those obtained from GPT-4 (Achiam et al., 2023) using the same prompts, which served as the ground truth for evaluation. The results provided in the Table 4 show that the generated response quality of both UA-CLM and CLM is similar for biography generation. Additionally, there is a significant improvement in calibration error and uncertainty quality, as quantified by hallucination detection AUROC, for UA-CLM.

## 5    DISCUSSION

We proposed a novel fine-tuning approach to improve uncertainty calibration in Large Language Models (LLMs) devised for natural language generation. Our method incorporates a differentiable uncertainty-aware causal language modeling loss, which is grounded in the principles of decision theory. This loss function is designed to enhance the model's ability to provide well-calibrated uncertainty estimates, a crucial aspect of trustworthy AI models.

Our extensive empirical evaluations on open-ended and free-form question-answering tasks has shown that the uncertainty-aware causal language modeling approach yield better-calibrated uncertainty quantification, which in turn significantly enhances the model's ability to detect hallucinations, identify out-of-domain prompts, and selective generation decisions. We demonstrated the generalizability of the proposed fine-tuning method to different text generation tasks. We also introduced a novel application of correlation analysis to the evaluation of sentence-level uncertainty calibration in free-form text generation, accounting for the varying sentence lengths between generated responses and ground-truth references.

**Limitations and Future work**: Currently, the proposed method is tailored to white-box model settings, where the model internals are accessible for calibration fine-tuning. However, there is a potential to extend uncertainty-aware fine-tuning to black-box models by calibrating an auxiliary model, or prompt tuning for calibrated uncertainty quantification. Additionally, the focus of this work has been on calibrating token-level uncertainty, which sets the stage for the exploration of calibrating sentence-level uncertainty. We plan to explore these two avenues in our future work. We hope this work opens new avenues for the research community to enhance the uncertainty calibration in LLMs for free-form natural language generation.

In conclusion, this work contributes towards the broader goal of developing trustworthy LLMs. The ability to recognize out-of-domain prompts and to acknowledge the limits of a model's knowledge base through reliable uncertainty quantification paves the way for reducing hallucinations and enhancing decision-making in AI systems.

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

ENHANCING TRUST IN LARGE LANGUAGE MODELS WITH
UNCERTAINTY-AWARE FINE-TUNING

## A APPENDIX

### A.1 EXPERIMENTAL DETAILS

#### A.1.1 DATASETS

**CoQA** Conversational Question Answering (CoQA) (Reddy et al., 2019) dataset was developed to evaluate models' ability to respond to natural, dialogue-based questions, with free-form text answers supported by highlighted evidence from the passage. The full dataset comprises of 127k question-answer pairs derived from 8k conversations based on text passages across 7 distinct domains. For all our experiments, we utilize the development subset of CoQA, which consists of 8k question-answer pairs. Figure 4 shows the color-coded co-reference chains in CoQA as illustrated in the (Reddy et al., 2019).

**TriviaQA** TriviaQA (Joshi et al., 2017) is a reading comprehension dataset consisting of over 650k question-answer-evidence triplets. It includes 95,000 question-answer pairs authored by trivia enthusiasts, along with an average of six independently gathered evidence documents per question, providing high-quality distant supervision for answering the questions. In our experiment, we used the validation split of the dataset with around 10,000 question-answer pairs. Table 5 shows some of the samples from the dataset.

**OK-VQA** Outside Knowledge-Visual Question Answering benchmarks (Marino et al., 2019) consists of visual queries where the image content alone is not sufficient to answer the questions. Thus, it requires models to incorporate external knowledge to generate accurate answers. The dataset consists of 14k questions across 10 knowledge categories. In our experiment, we used the validation split of the dataset with around 5k question-answer pairs. Figure 5 shows a few samples from the dataset across different knowledge categories.

---

The Virginia governor's race, billed as the marquee battle of an otherwise anticlimactic 2013 election cycle, is shaping up to be a foregone conclusion. Democrat Terry McAuliffe, the longtime political fixer and moneyman, hasn't trailed in a poll since May. Barring a political miracle, Republican Ken Cuccinelli will be delivering a concession speech on Tuesday evening in Richmond. In recent ...

$Q_1$: What are the candidates **running** for?
$A_1$: Governor
$R_1$: The Virginia governor's race

$Q_2$: **Where**?
$A_2$: Virginia
$R_2$: The Virginia governor's race

$Q_3$: Who is the democratic candidate?
$A_3$: **Terry McAuliffe**
$R_3$: Democrat Terry McAuliffe

$Q_4$: Who is **his** opponent?
$A_4$: **Ken Cuccinelli**
$R_4$ Republican Ken Cuccinelli

$Q_5$: What party does **he** belong to?
$A_5$: Republican
$R_5$: Republican Ken Cuccinelli

$Q_6$: Which of **them** is winning?
$A_6$: Terry McAuliffe
$R_6$: Democrat Terry McAuliffe, the longtime political fixer and moneyman, hasn't trailed in a poll since May

---

Figure 4: Sample from CoQA (Reddy et al., 2019) illustrating the co-reference chain of conversational questions.

| Question | Answer |
|---|---|
| Miami Beach in Florida borders which ocean? | Atlantic |
| What was the occupation of Lovely Rita according to the song by the Beatles | Traffic Warden |
| Who was Poopdeck Pappys most famous son? | Popeye |
| The Nazi regime was Germany's Third Reich; which was the first Reich? | HOLY ROMAN EMPIRE |

Table 5: Data samples from TriviaQA (Joshi et al., 2017)

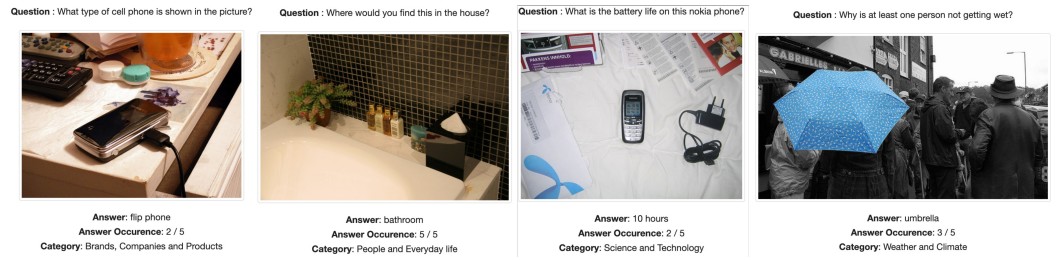

Figure 5: Data samples from OK-VQA (Marino et al., 2019) across different knowledge categories.

**BioASQ**    The BioASQ (Krithara et al., 2023) challenge, conducted every year, focuses on techniques in large-scale biomedical semantic indexing and question answering (QA). For our experiments, we utilize Task B (Table 6) from the eleventh edition of the BioASQ challenge (BioASQ 2023), which includes biomedical questions in English and their corresponding gold standard answers. We consider *exact answers* as gold answers where available; otherwise, we refer to the *ideal answers* field in the dataset.

| Question | Answer |
|---|---|
| Which amino acid in implicated in the Blue diaper syndrome? | tryptophan |
| What are the outcomes of ubiquitination? | Protein degradation, Degradation of proteins |
| What causes Serpentine Supravenous Hyperpigmentation? | 5-fluorouracil, docetaxel |
| What are positive cell-cycle regulators that can cause cancer when mutated called? | Proto-oncogenes |

Table 6: Data samples from BioASQ (Krithara et al., 2023)

### A.1.2    OPEN-BOOK QA PROMPT

Prompt:

> Answer the following question as briefly as possible.
> Context: [Provided context paragraph]
> Question: [Associated Question]
> Answer:

### A.1.3    FINETUNING HYPERPARAMETERS AND IMPLEMENTATION

We fine-tune our models for all experiments for 3 epochs using LoRA (Hu et al., 2022) with AdamW optmizer (Loshchilov & Hutter, 2019). We use an initial learning rate of $1e$-4, weight decay of $0.001$ and a warm up ratio of $0.03$. In our experiments we used Low-Rank Adaptation (LoRA) to efficiently fine-tune pre-trained LLMs and LVLMs for the causal language modeling task. For LLMs, we set the LoRA rank as 32, alpha parameter as 64 and a dropout of 0.1. LoRA was applied specifically to the following modules: *q_proj*, *k_proj*, *v_proj*, *up_proj*, and *down_proj*. In addition to LoRA, we applied 4-bit normalized float (*nf4*) quantization to the model's parameters and utilized *FP16* precision during fine-tuning to reduce the computational overhead.

For inference, we utilized *FP16* precision and the default greedy decoding provided by Hugging Face with temperature value T=0.3. The predictive entropy and semantic entropy are estimated by generating 5 stochastic sequences from the model, each obtained through temperature sampling with a temperature setting of T=0.3.This temperature was chosen to obtain optimal uncertainty estimates balanced with high quality generated text, based on the ablation study shown in Figure 6. Our source

code was implemented using Pytorch [1] framework and the models from Hugging Face [2] library. We will make the source code available to the community for reproducing the results.

For our LVLM model, LLaVA-1.5 (Liu et al., 2024a), we configured LoRA with a rank of 8, an alpha value of 8, and applied a 0.1 dropout rate to mitigate overfitting on the small OK-VQA training subset. In addition to the proposed UA-CLM loss, we experimented with a combined loss function that anneals the CLM loss with our UA-CLM loss. This approach allows the model to learn to answer OK-VQA queries using the context provided in the early stages of training, without uncertainty calibration. As training progresses, we shift our focus toward calibrating the model's uncertainty. By this stage, the model has already learned to answer visual question-answering prompts, allowing us to refine its performance on questions it is likely to answer correctly or incorrectly, based on insights gained during the initial training phases. Specifically, we assign a higher weight to the CLM loss in the early stages of training, gradually increasing the weight of the UA-CLM loss after 20% of the training is completed as shown in Equation 4. Our ablation results for this experiment are presented in Table 9.

$$\mathcal{L} = \mathcal{L}_{\text{CLM}} + \beta \cdot \mathcal{L}_{\text{UA-CLM}} \quad \text{where } \beta = \begin{cases} 0.2 & \text{if steps} \leq 0.2 \cdot \text{total\_steps} \\ 0.8 & \text{if steps} > 0.2 \cdot \text{total\_steps} \end{cases} \tag{4}$$

## A.2 TEXT GENERATION QUALITY METRICS

- **ROUGE-L (Lin & Och, 2004):** Recall-Oriented Understudy for Gisting Evaluation (ROUGE) is a widely-used evaluation metric for assessing the quality of text generated based on n-gram matching. We use the Rouge-L variant which uses the longest common subsequence between the generated answer and the ground truth answer.

- **Exact Match (EM):** Exact Match (EM) metric is a stringent evaluation criterion used to assess the performance of models on tasks such as question answering (QA), where a generated response is compared to a reference answer. It is a widely used metric for open-book QA, this metric evaluates a model's ability to extract the precise text span from the context to answer a question.

- **Accuracy:** The generated answer is considered as accurate if it achieves Rouge-L$(y, \hat{y}) > 0.3$, for a given reference answer $y$ and a model generation $\hat{y}$. We follow this criterion for quantifying accuracy in free-form text generation based on the findings from (Kuhn et al., 2023) that demonstrated this criterion closely matches the human evaluation accuracy on COQA and TriviaQA datasets, both of which are utilized in our experiments.

- **BERTScore (Zhang et al., 2020):** BERTScore utilizes word embeddings to compute a similarity score between the tokens in the prediction and ground truth and has shown to well correlate with human judgement. We report Precision, Recall and F1 BERTScores for all our experiments.

## A.3 UNCERTAINTY ESTIMATION METRICS

We assess uncertainty in natural language predictions by utilizing the Area Under the Receiver Operating Characteristic (AUROC) scores, calculated between correct and incorrect predictions across the following metrics:

- **Predictive Entropy** Fomicheva et al. (2020): This is a widely used measure for uncertainty estimation and is defined as the entropy of the model's output probability distribution from stochastic generated responses. Formally, for a specific instance $x$, the predictive entropy, denoted as $P_E(x)$, is defined as the conditional entropy of the output random variable $Y$, with realization $y$, given $x$ (Kuhn et al., 2023): $P_E(x) = H(Y|x) = -\int p(y|x) \ln p(y|x) dy$

- **Semantic Entropy** (Kuhn et al., 2023): Defined as entropy of output distributions in semantic event-space rather than traditional token event-space and has been shown to be a good indicator in detecting confabulation in language models.

---

[1]https://pytorch.org/

[2]https://huggingface.co/

Table 7: Evaluation of generated text quality metrics: Comparative analysis of Causal Language Modeling (CLM) and Uncertainty-aware Causal Language Modeling (UA-CLM) fine-tuning methods. The results in the table indicate that UA-CLM achievies similar or better generated text quality metrics than standard CLM across a range of models and datasets.

| Dataset | Model | Finetuning Method | Rouge-L | Exact Match | Accuracy | BERT Score (Precision) | BERT Score (Recall) | BERT Score (F1) |
|---|---|---|---|---|---|---|---|---|
| CoQA | Llama-2-7b | CLM | 0.8886 | 0.8071 | 0.9253 | 0.9633 | 0.9598 | 0.9604 |
| | | UA-CLM | 0.8882 | 0.8027 | 0.9264 | 0.9671 | 0.9644 | 0.9648 |
| | Llama-2-13b | CLM | 0.9106 | 0.8434 | 0.9406 | 0.9678 | 0.9639 | 0.9650 |
| | | UA-CLM | 0.9118 | 0.8204 | 0.9461 | 0.9732 | 0.9698 | 0.9705 |
| | Gemma-2b | CLM | 0.8654 | 0.7606 | 0.9143 | 0.962 | 0.9548 | 0.9570 |
| | | UA-CLM | 0.8632 | 0.7632 | 0.9088 | 0.9627 | 0.9554 | 0.9578 |
| TriviaQA | Llama-2-7b | CLM | 0.5867 | 0.4939 | 0.6385 | 0.8743 | 0.8785 | 0.8754 |
| | | UA-CLM | 0.6342 | 0.5627 | 0.6754 | 0.8951 | 0.8883 | 0.8910 |
| | Llama-2-13b | CLM | 0.6588 | 0.5883 | 0.6967 | 0.9026 | 0.8989 | 0.9001 |
| | | UA-CLM | 0.7277 | 0.6445 | 0.7710 | 0.9204 | 0.9164 | 0.9177 |
| | Gemma-2b | CLM | 0.4349 | 0.3674 | 0.4759 | 0.8375 | 0.8349 | 0.8355 |
| | | UA-CLM | 0.4563 | 0.3915 | 0.4959 | 0.8404 | 0.8382 | 0.8387 |
| OK-VQA | Llava-1.5-7b | CLM | 0.5569 | 0.5099 | 0.5891 | 0.8897 | 0.8864 | 0.8877 |
| | | UA-CLM | 0.5354 | 0.4950 | 0.5643 | 0.8841 | 0.8820 | 0.8827 |

Table 8: Uncertainty calibration analysis: The results show UA-CLM have more pronounced negative correlation between the uncertainty estimates and the generated text quality (ROUGE-L) than standard Causal Language Modeling CLM, indicating enhanced reliability in uncertainty quantification with UA-CLM.

| Dataset | Model | Finetuning Method | Spearman's rank correlation coefficient ↓ | | | | Pearson correlation coefficient ↓ | | | |
|---|---|---|---|---|---|---|---|---|---|---|
| | | | Token Entropy | Perplexity | Predictive Entropy | Semantic Entropy | Token Entropy | Perplexity | Predictive Entropy | Semantic Entropy |
| CoQA | Llama-2-7b | CLM | -0.2130 | -0.2379 | -0.3398 | -0.2898 | -0.2029 | -0.2109 | -0.2710 | -0.2881 |
| | | UA-CLM | **-0.2479** | **-0.3401** | **-0.4334** | **-0.3742** | **-0.3414** | **-0.3414** | **-0.3414** | **-0.3414** |
| | Llama-2-13b | CLM | -0.2325 | -0.2523 | -0.3253 | -0.3004 | -0.2302 | -0.2495 | -0.3001 | -0.2636 |
| | | UA-CLM | **-0.2398** | **-0.3280** | **-0.4170** | **-0.3717** | **-0.2335** | **-0.3244** | **-0.3269** | **-0.3481** |
| | Gemma-2b | CLM | -0.3639 | -0.3629 | -0.4335 | -0.3756 | -0.3860 | -0.3713 | -0.3483 | -0.3399 |
| | | UA-CLM | **-0.3676** | **-0.4063** | **-0.4476** | **-0.4127** | **-0.4033** | **-0.4019** | **-0.3517** | **-0.3530** |
| TriviaQA | Llama-2-7b | CLM | -0.5627 | -0.5863 | -0.5765 | -0.5994 | -0.5047 | -0.4854 | -0.2864 | -0.5020 |
| | | UA-CLM | **-0.5713** | **-0.6011** | **-0.5822** | -0.5980 | **-0.5385** | **-0.5326** | **-0.3382** | -0.4916 |
| | Llama-2-13b | CLM | -0.5711 | -0.5845 | -0.5522 | -0.5959 | -0.5155 | -0.4915 | -0.4548 | -0.4612 |
| | | UA-CLM | **-0.5725** | **-0.5862** | **-0.5607** | -0.5854 | **-0.5362** | **-0.5407** | **-0.4786** | -0.4479 |
| | Gemma-2b | CLM | -0.5636 | -0.5772 | -0.5609 | -0.5537 | -0.5020 | -0.4534 | -0.4494 | -0.4514 |
| | | UA-CLM | -0.5623 | **-0.5913** | -0.5457 | **-0.5928** | **-0.5164** | **-0.5010** | **-0.4534** | **-0.4947** |
| OK-VQA | Llava-1.5-7b | CLM | -0.1253 | -0.1132 | -0.1320 | -0.1062 | -0.0862 | -0.0861 | -0.1256 | -0.1340 |
| | | UA-CLM | **-0.1606** | **-0.1619** | **-0.2050** | **-0.2660** | -0.0748 | **-0.1214** | **-0.2100** | **-0.3020** |

- **Perplexity** Fomicheva et al. (2020): A standard metric to assess the quality of model and is defined as the inverse probability of the generated text: Perplexity $= \exp\left(-\frac{1}{N}\sum_{i=1}^{N}\log_2 p(w_i|w_1,\ldots,w_{i-1})\right)$

## A.4 ADDITIONAL RESULTS

The results in the Table 7 presents a detailed quantitative evaluation of various text generation quality metrics across various models, datasets, and uncertainty quantification (UQ) metrics. It compares standard Causal Language Modeling (CLM) with our Uncertainty-Aware Causal Language Modeling (UA-CLM).

The results in Table 8 presents quantitative data with the values of Spearman's rank correlation coefficient and Pearson correlation coefficient across different models, datasets, and uncertainty quantification (UQ) metrics, with a specific focus on comparing standard Causal Language Modeling (CLM) and our Uncertainty-Aware Causal Language Modeling (UA-CLM). The data reveals that UA-CLM exhibits a stronger inverse correlation between UQ metrics and ROUGE-L scores, indi-

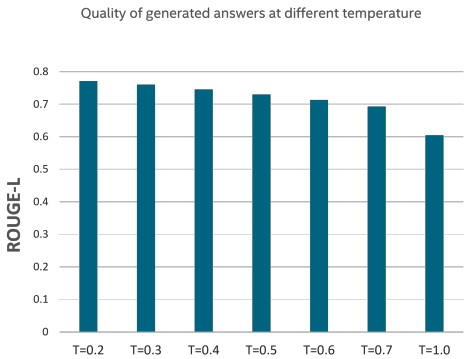 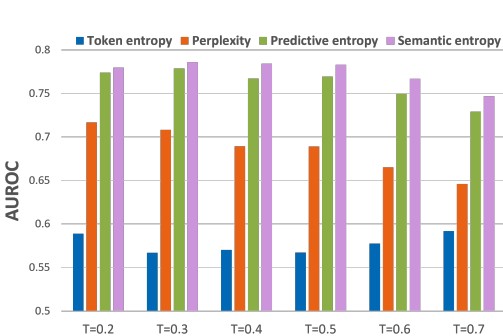

Figure 6: Ablation study: Effect of temperature value on the quality of generated text and the quality of uncertainty estimates evaluated with AUROC for hallucination detection. The study was performed on pre-trained Llama-2-7B model with CoQA dataset. Based on this study, we selected temperature T=0.3 as it results in optimal AUROC and ROUGE-L scores.

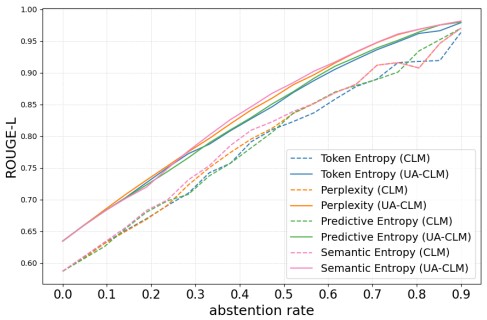 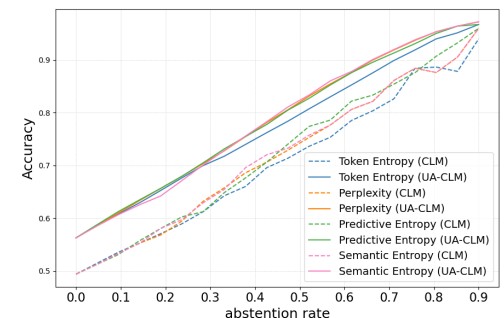

Figure 7: Selective generation (Llama-2-7B/TriviaQA)

cating better reliability of uncertainty estimates. This enhanced inverse relationship suggests that UA-CLM is more adept at associating higher uncertainty with low quality text generation quality and vice versa, which is a key indicator of better uncertainty calibration.

Table 9: Ablation study: Effect of different loss functions during fine-tuning. Exact match is used as accuracy metric in computing AUARC.

| Dataset | Model | Fine-tuning Loss | AUROC (Hallucination/Confabulation detection) | | | | AUARC (Area under rejection accuracy curve) | | | |
|---|---|---|---|---|---|---|---|---|---|---|
| | | | Token Entropy | Perplexity | Predictive Entropy | Semantic Entropy | Token Entropy | Perplexity | Predictive Entropy | Semantic Entropy |
| OKVQA | Llava-1.5-7b | $\mathcal{L}_{\text{CLM}}$ | 0.5504 | 0.5419 | 0.5455 | 0.537 | 0.5809 | 0.5781 | 0.579 | 0.5747 |
| | | $\mathcal{L}_{\text{UA-CLM}}$ | 0.5839 | 0.6032 | 0.5701 | 0.6727 | 0.5657 | 0.5771 | 0.5601 | 0.6028 |
| | | $\mathcal{L}_{\text{CLM}} + \beta * \mathcal{L}_{\text{UA-CLM}}$ | 0.6001 | 0.5984 | 0.6106 | 0.6638 | 0.5989 | 0.5965 | 0.6012 | 0.6265 |
| CoQA | Llama-2-7b | $\mathcal{L}_{\text{CLM}}$ | 0.6252 | 0.632 | 0.6635 | 0.6889 | 0.823 | 0.829 | 0.8516 | 0.8405 |
| | | $\mathcal{L}_{\text{UA-CLM}}$ | 0.6955 | 0.7398 | 0.7413 | 0.7741 | 0.8246 | 0.8477 | 0.8743 | 0.8571 |
| | | $\mathcal{L}_{\text{CLM}} + \beta * \mathcal{L}_{\text{UA-CLM}}$ | 0.6101 | 0.6183 | 0.6978 | 0.7252 | 0.8153 | 0.8153 | 0.8614 | 0.8455 |
| TriviaQA | Llama-2-13b | $\mathcal{L}_{\text{CLM}}$ | 0.8264 | 0.8333 | 0.7971 | 0.8407 | 0.7464 | 0.7526 | 0.7532 | 0.7556 |
| | | $\mathcal{L}_{\text{UA-CLM}}$ | 0.8297 | 0.8352 | 0.8033 | 0.8447 | 0.7960 | 0.8059 | 0.804 | 0.8069 |
| | | $\mathcal{L}_{\text{CLM}} + \beta * \mathcal{L}_{\text{UA-CLM}}$ | 0.8340 | 0.8263 | 0.8049 | 0.8307 | 0.7666 | 0.7692 | 0.7673 | 0.7693 |

Figure 7 shows results on selective generation, based on varying levels of abstaining from providing generated response informed by uncertainty estimates. We plotted both ROUGE-L scores and accuracy as functions of the abstention rate, showing how the models perform as they increasingly withhold responses in situations of high uncertainty. The plots clearly shows that the UA-CLM outperforms CLM across all the four uncertainty metrics.

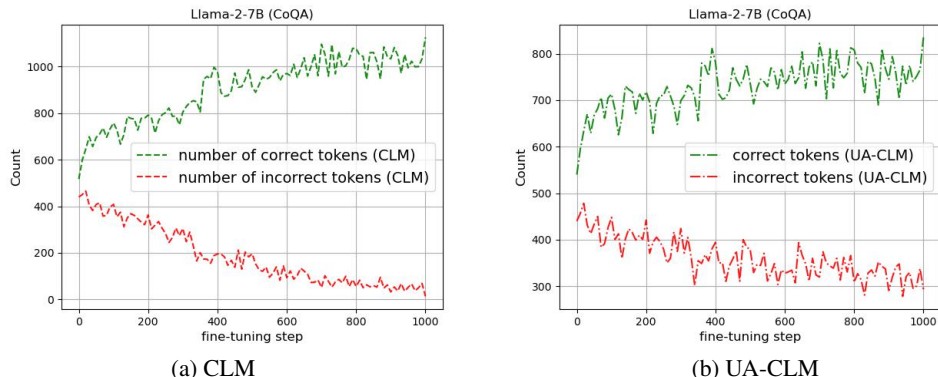

(a) CLM                 (b) UA-CLM

Figure 8: Analysis of Correct and Incorrect Token Counts in mini-batch during fine-tuning with CLM and UA-CLM. Both CLM and UA-CLM show increase in correct tokens and a decrease in incorrect tokens as fine-tuning progresses.

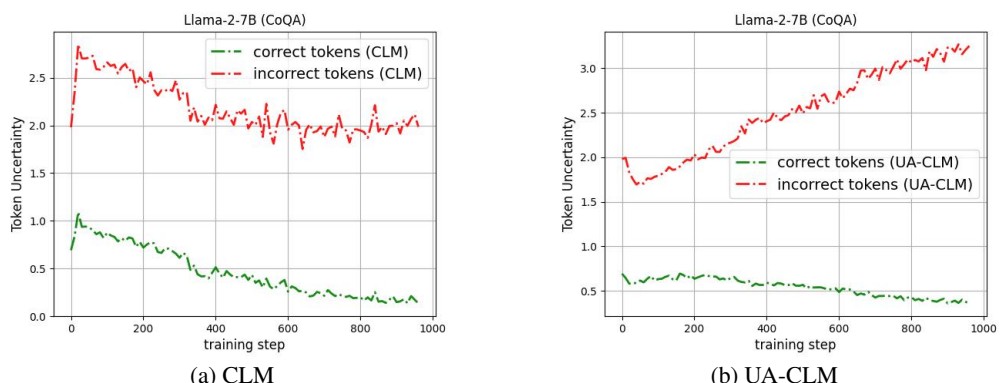

(a) CLM                 (b) UA-CLM

Figure 9: Analysis of Token Uncertainty associated with Correct and Incorrect tokens in the mini-batch during fine-tuning with CLM and UA-CLM. A well-calibrated model should provide low uncertainty for correct tokens and higher uncertainty for incorrect tokens. With standard CLM Loss, uncertainty for both correct and incorrect tokens decreases, indicating overconfidence even on incorrect tokens. In contract, with UA-CLM, the uncertainty for incorrect tokens increases and the decreasing uncertainty on correct tokens, supporting that the fine-tuning with UA-CLM improves the reliability of uncertainty estimates.

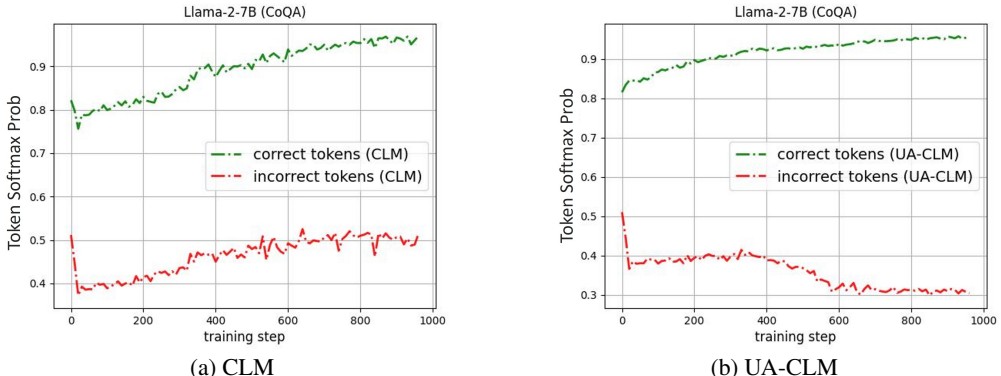

(a) CLM                 (b) UA-CLM

Figure 10: Analysis of Token Softmax Probability associated with Correct and Incorrect tokens during fine-tuning with CLM and UA-CLM. A well-calibrated model should assign high probability to correct tokens and lower probability to incorrect tokens. With standard CLM loss, probabilities for both correct and incorrect tokens increase as fine-tuning progress, indicating overconfidence. In contrast, UA-CLM fine-tuning results in higher probabilities for correct tokens and lower probabilities for incorrect tokens, enhancing the reliability of token probability scores

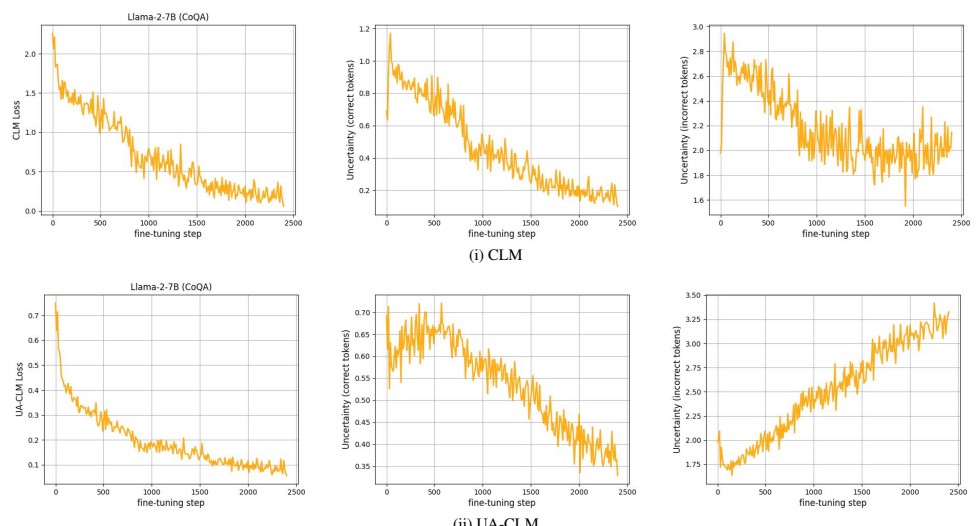

Figure 11: Llama-2-7B: Loss convergence and uncertainty values associated with correct and incorrect tokens.

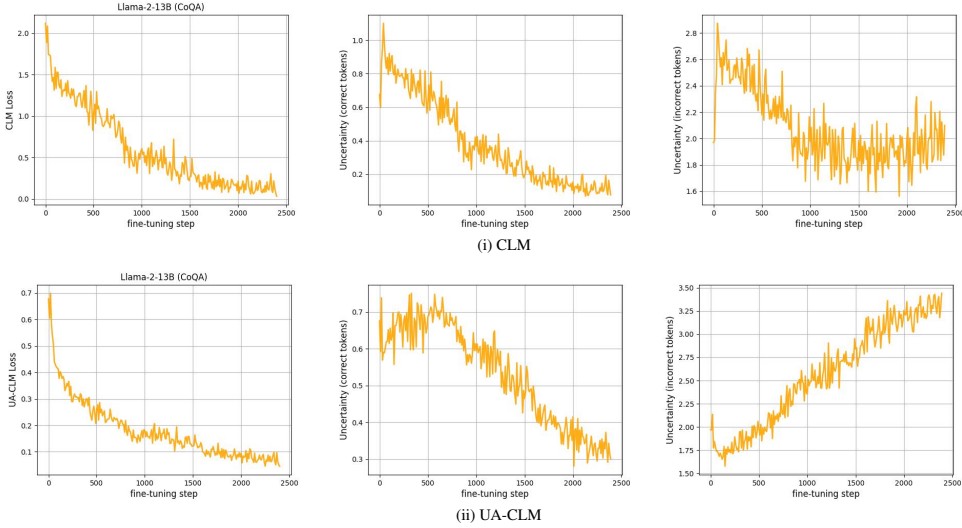

Figure 12: Llama-2-13B: Loss convergence and uncertainty values for correct and incorrect tokens.

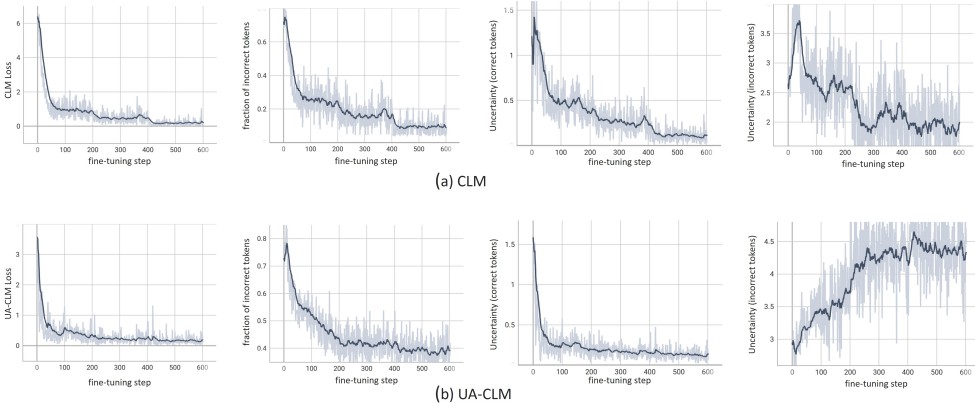

Figure 13: Llava-1.5: Loss convergence and uncertainty values associated with correct and incorrect tokens.

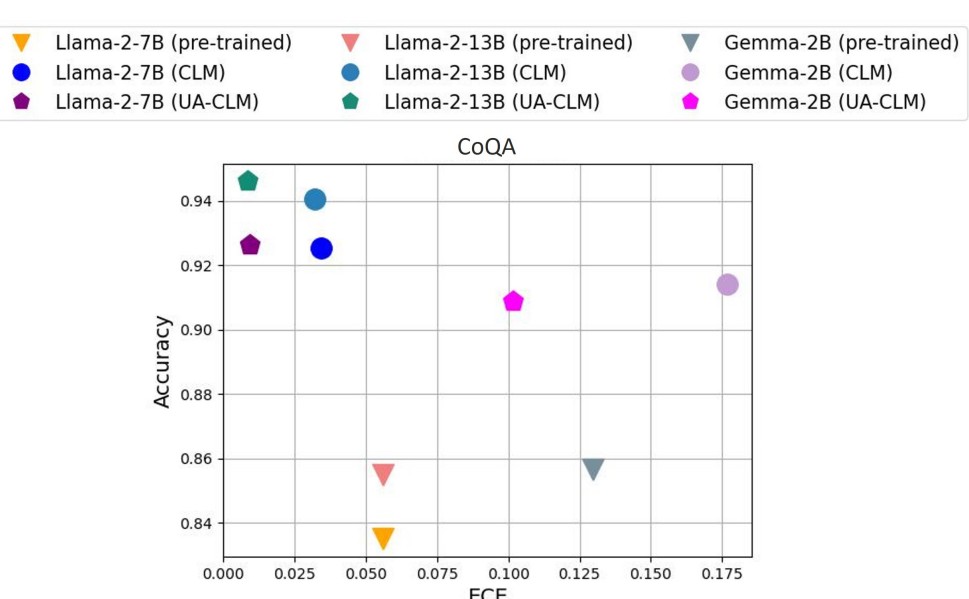

Figure 14: Accuracy versus Expected Calibration Error (ECE) comparison between UA-CLM, CLM, and pre-trained baseline across different LLM architectures on CoQA dataset. The ideal model should have high accuracy and low expected calibration error, indicating accurate predictions with well-calibrated uncertainty quantification (top-left of the Accuracy vs ECE plot). When evaluating three different model architectures, we observe that the accuracy of models with CLM and UA-CLM remains within a similar range and better than the pre-trained baseline. While, the ECE of models fine-tuned with UA-CLM shows significant improvement compared to both the pre-trained baseline and CLM fine-tuning.

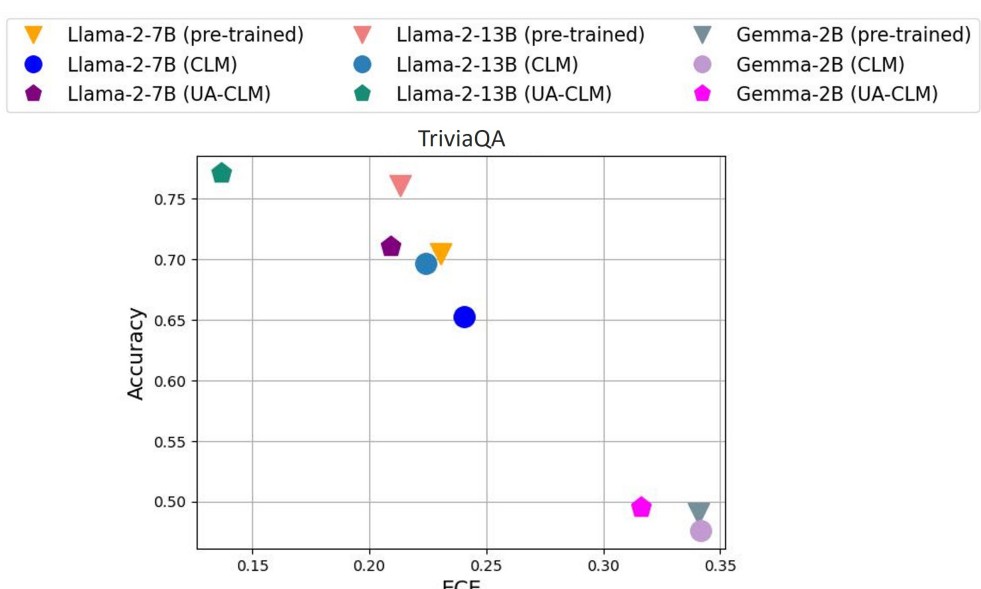

Figure 15: Accuracy versus Expected Calibration Error (ECE) comparison between UA-CLM, CLM, and pre-trained baseline across different LLM architectures on TriviaQA dataset. The ideal model should have high accuracy and low expected calibration error, indicating accurate predictions with well-calibrated uncertainty quantification (top-left of the Accuracy vs ECE plot). When evaluating three different model architectures, we observe that the both accuracy and ECE of the models fine-tuned with UA-CLM shows significant improvement compared to both the pre-trained baseline and CLM fine-tuning.

