# OpenReview forum: "Enhancing Trust in Large Language Models with Uncertainty-Aware Fine-Tuning"
_ICLR.cc/2025/Conference — ICLR 2025 Conference Withdrawn Submission_

### Official Review · Reviewer_wXAr · 2024-11-04

**Soundness:** 3
**Presentation:** 4
**Contribution:** 3
**Rating:** 6
**Confidence:** 4

**Summary:**

The paper proposes a new approach to align the confidence and accuracy of causal (auto-regressive) large language models. This approach is based on low-rank fine-tuning, adding a component to the fine-tuning loss that incorporates not only the auto-regressive negative log likelihood but also aligns token entropy with prediction accuracy. In other words, this new objective for fine-tuning large language models aligns their prediction accuracy with the uncertainty. The paper presents experiments using various LLMs, such as Llama and Gemma, and evaluates the approach on tasks like hallucination detection, selective generation, out-of-prompt detection, and visual question answering. The results show effective compared to baseline LLMs.

**Strengths:**

The paper is written very clearly.
The related literature review is thorough and nicely structured.
The proposed loss and the corresponding optimization with two objectives of optimizing accuracy as well as aligning the accuracy with confidence is a sound and interesting idea.
The results compared to the baseline LLMs on the datasets and with selected metrics show very effective.

**Weaknesses:**

--While the approach looks effective empirically, the results have not been compared to the related work. For example some of your QA benchmarks overlap with [2] but there is no comparisons made. This makes it hard to see how the proposed technique is compared to other existing techniques for the same purpose.  Can you clarify further on this issue?

--Can you clarify, why the expected calibration error ECE is not reported? [see this paper that reports ECE [1]]

--It will be useful to define the evaluation metrics, their motivation, and how those are computed. This will make the paper self-contained for its major concepts.
--Related to the above question, it wasn’t clear to me how we would solicit the model's uncertainty for each instance, for example,  in the QA setting? do you compute H?

[1] @misc{huang2024verbalizedprobabilisticgraphicalmodeling,
      title={Verbalized Probabilistic Graphical Modeling with Large Language Models},
      author={Hengguan Huang and Xing Shen and Songtao Wang and Dianbo Liu and Hao Wang},
      year={2024},
      eprint={2406.05516},
      archivePrefix={arXiv},
      primaryClass={cs.LG},
      url={https://arxiv.org/abs/2406.05516},
}
[2] @misc{lin2024generatingconfidenceuncertaintyquantification,
      title={Generating with Confidence: Uncertainty Quantification for Black-box Large Language Models},
      author={Zhen Lin and Shubhendu Trivedi and Jimeng Sun},
      year={2024},
      eprint={2305.19187},
      archivePrefix={arXiv},
      primaryClass={cs.CL},
      url={https://arxiv.org/abs/2305.19187},
}

**Questions:**

See above.

---

> ### Author Response · Authors · 2024-11-19
> **Author Response to Reviewer wXAr (Part 1)**
>
> Dear Reviewer wXAr,
>
> Thank you for your thoughtful review and feedback. We appreciate your recognition that our paper is written very clearly, thoroughness of related literature review, soundness of our proposed loss and optimization objectives, and the effectiveness of experimental results. We value your constructive feedback and would like to address your concerns to improve our paper further.
>
> >**[W1]:** *"While the approach looks effective empirically, the results have not been compared to the related work. For example some of your QA benchmarks overlap with [2] but there is no comparisons made....Can you clarify further on this issue?"*
>
> The uncertainty metric *"Eccentricity"* introduced in [2] (https://arxiv.org/abs/2305.19187) is a black-box uncertainty estimation technique that relies solely on the diversity of multiple generated text sequences and does not depend on the likelihood scores of the generated tokens in the sentence. Since our proposed technique directly optimizes the likelihood scores of the generated tokens, we focused on related uncertainty metrics such as token entropy, perplexity, predictive entropy, and semantic entropy in our study. We have now compared the *Eccentricity* along with the entropy-based uncertainty metrics. The results of this comparison are presented in the table below.
>
> Additionally, in regard to related works on fine-tuning, we have compared to Calibration Tuning (CT) [Kapoor et al. (2024)] and Unlikelihood training [Welleck et al. (2020)] methods, as suggested by Reviewer gRZf. These comparisons are presented in the table above in response [W2] to Reviewer gRZf. We will include these comparisons in the revised manuscript to provide a more comprehensive comparison of our approach with related works.
>
> | Method/Model    |   | AUROC ↑ (Hallucination detection)  |            |                    |                  |               |   | AUARC ↑ (Selective generation Accuracy) |            |                    |                  |               |   | AUARC ↑ (Selective generation Exact Match) |            |                    |                  |               |
> |-----------------|---|------------------------------------|------------|--------------------|------------------|---------------|---|-----------------------------------------|------------|--------------------|------------------|---------------|---|--------------------------------------------|------------|--------------------|------------------|---------------|
> |                 |   |            Token Entropy           | Perplexity | Predictive Entropy | Semantic Entropy | **Eccentricity**  |   |              Token Entropy              | Perplexity | Predictive Entropy | Semantic Entropy | **Eccentricity**  |   |                Token Entropy               | Perplexity | Predictive Entropy | Semantic Entropy | **Eccentricity**  |
> | **Llama-2-7B** |   |                                    |            |                    |                  |               |   |                                         |            |                    |                  |               |   |                                            |            |                    |                  |               |
> | CLM             |   |               0.6252               |   0.6320   |       0.6635       |      0.6889      |     0.6845    |   |                  0.9435                 |   0.9444   |       0.9508       |      0.9530      |     0.9526    |   |                   0.8230                   |   0.8290   |       0.8516       |      0.8405      |     0.8238    |
> | UA-CLM (ours)   |   |             **0.6955**             | **0.7398** |     **0.7413**     |    **0.7741**    |   **0.6957**  |   |                **0.9603**               | **0.9657** |     **0.9699**     |    **0.9716**    |   **0.9545**  |   |                 **0.8246**                 | **0.8477** |     **0.8743**     |    **0.8571**    |   **0.8261**  |
> |                 |   |                                    |            |                    |                  |               |   |                                         |            |                    |                  |               |   |                                            |            |                    |                  |               |
>
> | Method        | AUROC (Out-of-domain prompt detection) ↑ |               |            |                    |                  |
> |---------------|-------------------------------------|---------------|------------|--------------------|------------------|
> |             |           **Eccentricity**          | Token entropy | Perplexity | Predictive Entropy | Semantic Entropy |
> | CLM           |                0.7862               |     0.7828    |   0.7661   |        0.75        |      0.7902      |
> | UA-CLM (ours) |              **0.8928**             |   **0.9025**  | **0.8931** |     **0.7763**     |    **0.9061**    |
> response continued below...

---

> ### Author Response · Authors · 2024-11-19
> **Author Response to Reviewer wXAr (Part 2)**
>
> > **[W2]:** *"Can you clarify, why the expected calibration error ECE is not reported? [see this paper that reports ECE [1]]"*
>
> Thank you for your insightful question, suggestion, and for sharing the reference. Initially, we did not report ECE because it is not feasible to compute token-level ECE due to potential mismatches in the number of tokens between the generated text and the ground truth reference. However, sentence-level ECE can be computed based on the accuracy of the generated sentence with respect to the ground truth, similar to the paper you referenced [1]. We have now computed ECE, the results are presented in the Table below, we can observe that our method UA-CLM yields lower Expected Calibration Error. We will report these findings in the revised manuscript.
>
> | Model       | Method                    |   | ECE ↓      |   |            |
> |-------------|---------------------------|---|------------|---|------------|
> |             |                           |   |    CoQA    |   |  TriviaQA  |
> |             |                           |   |            |   |            |
> | Llama-2-7B  | Pretrained                |   |   0.0561   |   |   0.2304   |
> |             | CLM                       |   |   0.0343   |   |   0.2407   |
> |             | ULT [Welleck et al. 2020] |   |   0.0588   |   |   0.3111   |
> |             | CT  [Kapoor et al. 2024]  |   |    0.0410   |   |   0.2276   |
> |             | UA-CLM (ours)             |   | **0.0094** |   |  **0.2090** |
> |             |                           |   |            |   |            |
> | Llama-2-13B | Pretrained                |   |   0.0559   |   |   0.2133   |
> |             | CLM                       |   |   0.0323   |   |   0.2241   |
> |             | ULT [Welleck et al. 2020] |   |   0.0595   |   |   0.1517   |
> |             | CT  [Kapoor et al. 2024]  |   |   0.0992   |   |   0.1937   |
> |             | UA-CLM (ours)             |   | **0.0084** |   | **0.1365** |
> |              |                                     |    |                  |    |                 |
>
> Additionally, we have performed the **Accuracy versus ECE trade-off analysis** across different models for UA-CLM and CLM, we have presented the findings in **Figures 13 and 14 in the Appendix (page 21)** of the revised manuscript.
>
> > **[W3]:** *"It will be useful to define the evaluation metrics, their motivation, and how those are computed."*
>
> Thank you for your suggestion. We agree and will include detailed definitions of the evaluation metrics in the revised manuscript, in addition to the information already provided in Appendix A.3 (supplementary material).
>
> >**[Q]:** *"Related to the above question, it wasn’t clear to me how we would solicit the model's uncertainty for each instance, for example, in the QA setting? do you compute H?"*
>
> We use the uncertainty metrics from existing literature - token entropy [Fomicheva et al. 2020], predictive entropy [Malinin and Gales 2021], semantic entropy [Kuhn et al. 2023]. To clarify your question on how we solicit the model's uncertainty for each instance in the QA setting, we compute the entropy (H) of the predicted probability distribution for each generated token across the vocabulary size and then calculate the mean across all the tokens in the sentence to obtain the mean token entropy for the generated response in the QA setting.  For predictive entropy,  we compute the entropy of the model’s output probability distribution from multiple stochastic generated text responses.  For semantic entropy, the multiple stochastic generated sentences are first clustered together based on their semantic meaning, and then compute the entropy of the resulting probability distribution from clustering. We will include more clarity on the uncertainty metrics computation techniques in the revised manuscript.
>
> We hope that our responses and additional experiments adequately address your concerns and questions. If you find that our responses satisfactorily address your feedback, we kindly request that you consider upgrading your evaluation.
>
> Ref:
>
> [Welleck et al. (2020)] Welleck, S., Kulikov, I., Roller, S., Dinan, E., Cho, K. and Weston, J., Neural Text Generation With Unlikelihood Training. ICLR 2020.
>
> [Kapoor et al. (2024)] Kapoor, Sanyam, Nate Gruver, Manley Roberts, Arka Pal, Samuel Dooley, Micah Goldblum, and Andrew Wilson. "Calibration-Tuning: Teaching Large Language Models to Know What They Don’t Know." In Proceedings of the 1st Workshop on Uncertainty-Aware NLP (UncertaiNLP 2024).

---

> > ### Comment · Reviewer_wXAr · 2024-11-24
> >
> > Thank you for your response to all my questions and comments. I am reviewing all the responses and will update my scores if applicable. I have no more questions for now.  Thanks.

---

> ### Author Response · Authors · 2024-12-02
>
> Dear Reviewer **wXAr**,
>
> Thank you for your prompt response. Your comments and suggestions are very valuable to us. Please let us know if our rebuttal has satisfactorily addressed your comments and questions. We would be grateful if you could consider updating your evaluation if our responses and additional experimental results has addressed your concerns.

---

### Official Review · Reviewer_Br54 · 2024-11-04

**Soundness:** 3
**Presentation:** 3
**Contribution:** 3
**Rating:** 6
**Confidence:** 3

**Summary:**

This paper proposes uncertainty-aware fine-tuning as a method to increase the reliability of existing uncertainty quantification metrics. The method is simple, unlike negative log-likelihood (or cross-entropy), by distinguishing correct tokens and incorrect tokens and adding a loss term for entropy in the optimization objective, it can lower predictive uncertainty in the case of correct tokens and vice versa. With this objective, language models and even vision language models are fine-tuned and compared with standard fine-tuned models, on existing entropy based uncertainty metrics, such as token entropy, perplexity, predictive entropy and semantic entropy.

**Strengths:**

By designing uncertainty loss based on decision theory, it ensures the loss can theoretically converge to 0, leading to improved performance of general entropy-based metrics without compromising accuracy.
The authors provide rigorous evaluations across multiple question-answering datasets, showcasing the method’s efficacy in various contexts. These experimental results demonstrate that the uncertainty-aware fine-tuning improves in four aspects, hallucination detection, selective generation, out-of-prompt detection and calibration analysis.

**Weaknesses:**

The authors said that (Rouge-L >0.3) was used to measure accuracy as same as (Khun et al., 2023). However, as also pointed out in the previous work, the Rouge-L score is a very brittle metric, and unlike the previous work that only draw curves according to temperature, this paper needs to capture the trade-off between accuracy and uncertainty calibration. As shown in Table 2, we can see that the accuracy of UA-CLM is higher than that of CLM except for Gemma, which is quite weird because the authors mention the trade-off. The reason for this is not adequately explained.

**Questions:**

Are there any experimental results conducted using standard fine-tuning other than LoRA? (or any other PEFT methods, such as prompt tuning.)

Looking at the loss term alone, there seems to be problems like longer training time or faster loss convergence. Are there any reports on these?

---

> ### Author Response · Authors · 2024-11-18
> **Author Response to Reviewer Br54 (Part 1)**
>
> Dear Reviewer Br54,
>
> Thank you for your insightful review and feedback. We are glad that you found our proposed uncertainty loss based on decision theory to be theoretically sound and effective in improving general entropy-based metrics. We also appreciate your recognition of our rigorous experimental evaluation and efficacy in hallucination detection, selective generation, out-of-prompt detection and calibration analysis. We address your concerns and questions below.
>
> > **[W1]:** *"The authors said that (Rouge-L >0.3) was used to measure accuracy as same as (Khun et al., 2023). However, as also pointed out in the previous work, the Rouge-L score is a very brittle metric.... "*
>
> We agree that selecting an appropriate accuracy evaluation metric for free-form text generation is both critical and challenging. In our study, we followed the findings from prior work by [Kuhn et al. (2023)] in choosing the Rouge-L criterion for accuracy. Their work had manually evaluated and verified that Rouge-L > 0.3 closely matches human evaluations on the same datasets we are using (both CoQA and TriviaQA). Additionally, we also considered the Exact Match criterion and observed similar AUROC performance on CoQA and TriviaQA tasks, as these involve phrase or sentence-length generations (we had provided the Exact Match values in Appendix Table 6). However, since Exact match may not be ideal metric for longer sentences, we opted to use Rouge-L based criterion on the supporting data from [Kuhn et al. (2023)] with human evaluations.
>
> > **[W1]:** *"....this paper needs to capture the trade-off between accuracy and uncertainty calibration..."*
>
> Thank you for the suggestion. We followed your advice and performed a study on the trade-off between accuracy and calibration error. The results of this study have been included in **Figures 14 and 15 in the Appendix (page 22)** of the revised manuscript.
>
> In the accuracy versus expected calibration error (ECE) plot for CoQA dataset, we observe that the accuracy of models fine-tuned with both CLM and UA-CLM remains within a similar range and is better than the pre-trained baseline. While, the ECE of models fine-tuned with UA-CLM shows significant improvement compared to both the pre-trained baseline and CLM fine-tuning. For the TriviaQA dataset, we observe that both the accuracy and ECE of models fine-tuned with UA-CLM show improvement compared to both the pre-trained baseline and CLM fine-tuning.
>
> > **[W1]:** *"... we can see that the accuracy of UA-CLM is higher than that of CLM except for Gemma, which is quite weird because the authors mention the trade-off."*
>
> Thank you for your observation. This behavior appears to be more dataset-dependent rather than model-dependent. As shown in the accuracy versus ECE plots, the accuracy remains similar for both CLM and UA-CLM finetuned models on the CoQA dataset, while there is an improvement in accuracy for TriviaQA, along with a reduction in calibration error. Further, our proposed optimization objective is designed to encourage better uncertainty calibration while still maximizing accuracy within the framework of causal language modeling. We present the accuracy and ECE values in the table below for easier comparison.
>
> | Model           | CoQA           |            |   | TriviaQA       |            |
> |-----------------|----------------|------------|---|----------------|------------|
> |                 | **Accuracy ↑** |  **ECE ↓** |   | **Accuracy ↑** |  **ECE ↓** |
> | **Llama-2-7B**  |                |            |   |                |            |
> |                 |                |            |   |                |            |
> | CLM             |     0.9253     |   0.0343   |   |     0.6529     |   0.2407   |
> | UA-CLM (ours)          |   **0.9264**   | **0.0094** |   |   **0.7108**   | **0.2090** |
> |                 |                |            |   |                |            |
> | **Llama-2-13B** |                |            |   |                |            |
> |                 |                |            |   |                |            |
> | CLM             |     0.9406     |   0.0323   |   |     0.6970     |   0.2241   |
> | UA-CLM (ours)          |   **0.9461**   | **0.0084** |   |   **0.7710**   | **0.1365** |
> |                 |                |            |   |                |            |
> | **Gemma-2B**    |                |            |   |                |            |
> |                 |                |            |   |                |            |
> | CLM             |   **0.9143**   |   0.2241   |   |     0.4759     |   0.3418   |
> | UA-CLM (ours)          |     0.9088     | **0.1365** |   |   **0.4959**   | **0.3163** |
> |                 |                |            |   |                |            |
>
> Ref:
>
> [Kuhn et al. (2023)] Kuhn, Lorenz, Yarin Gal, and Sebastian Farquhar. "Semantic Uncertainty: Linguistic Invariances for Uncertainty Estimation in Natural Language Generation." ICLR 2023.

---

> ### Author Response · Authors · 2024-11-18
> **Author Response to Reviewer Br54 (Part 2)**
>
> >**[Q1]:** *"Are there any experimental results conducted using standard fine-tuning other than LoRA? (or any other PEFT methods, such as prompt tuning.)"*
>
> We have primarily focused on LoRA for fine-tuning in this study. As mentioned in the third paragraph of the discussion section of the manuscript, this work concentrated on white-box model settings. There is potential to extend uncertainty-aware fine-tuning to black-box models by calibrating an auxiliary model or using prompt tuning for calibrated uncertainty quantification, which we plan to explore in future works.
>
> The design choice of using LoRA for parameter-efficient fine-tuning (PEFT) in white-box settings is further supported by findings from recent work [Wistuba et al. (2024)], which demonstrate that LoRA significantly outperforms prompt tuning in white-box PEFT setups.
>
> >**[Q2]:** *"Looking at the loss term alone, there seems to be problems like longer training time or faster loss convergence. Are there any reports on these?"*
>
> Thank you for the question, these are valid points. Please refer to the table below for the training latency and throughput comparison. Our method has similar training time as standard CLM finetuning with benefits of improved uncertainty calibration. Latency measurements for both methods were conducted under similar conditions for a fair comparison: a batch size of 2 with an average of 512 tokens per sequence, 4-bit normalized float (nf4) quantization for model parameters, and FP16 precision for computation on Intel(R) Xeon(R) Gold 6448H CPU and Nvidia A100 80G GPU, with LoRA fine-tuning (less than 1% model parameters updated), as described in Appendix A.1.3. While the numbers may vary with different setups, the relative comparison should remain consistent.
>
> Following your suggestion, we have included loss convergence plots and detailed information on the uncertainty values associated with correct and incorrect tokens during the fine-tuning process in the revised manuscript (see **Appendix Figures 11-13 in page 21**). Additional analysis and findings can be found in **Appendix Figures 8-10 in page 20**.  The findings as the loss converges are as follows: With the standard CLM loss, uncertainty decreases for both correct and incorrect tokens, indicating overconfidence even in incorrect tokens. In contrast, with UA-CLM, uncertainty for incorrect tokens increases while uncertainty for correct tokens decreases. This supports the conclusion that fine-tuning with UA-CLM improves the reliability of uncertainty estimates.
>
> | Method    |   Latency ↓   | Throughput ↑ | Latency difference |
> |-----------------|:-------------:|:------------:|:------------------:|
> | **Llama-2-7B**  |               |              |                   |
> | CLM             | 2.63 ms/token | 379 token/s  |        1x            |
> | UA-CLM (ours)         | 2.78 ms/token | 359 token/s  | 1.06x          |
> |                 |               |              |                    |
> | **Llama-2-13B** |               |              |                    |
> | CLM             | 4.50 ms/token | 222 token/s  |         1x           |
> | UA-CLM (ours)          | 4.85 ms/token | 206 token/s  | 1.07x          |
>
>
> Thank you again for your valuable feedback, which helps us in improving the quality of our work. Please let us know if you have any further questions. If our responses address your concerns, we kindly request that you consider updating your evaluation.
>
> Ref:
>
> [Wistuba et al. 2024] Wistuba, Martin, Prabhu Teja Sivaprasad, Lukas Balles, and Giovanni Zappella. "Choice of PEFT Technique in Continual Learning: Prompt Tuning is Not All You Need." arXiv preprint arXiv:2406.03216 (2024).

---

> > ### Author Response · Authors · 2024-12-02
> >
> > Dear Reviewer **Br54**,
> >
> > We have made every effort to address all your concerns and questions in our responses above. We kindly request that you review our rebuttal and let us know if you have any additional questions. If you find that our rebuttal has satisfactorily addressed your concerns, we would be grateful if you could consider updating your evaluation. Thank you once again for your time to review our work and we look forward to hearing back from you soon.

---

### Official Review · Reviewer_RnpC · 2024-11-06

**Soundness:** 2
**Presentation:** 3
**Contribution:** 2
**Rating:** 3
**Confidence:** 4

**Summary:**

This work proposes a fine-tuning approach to better calibrate language models with their uncertainty levels. Specifically, they introduce an uncertainty-aware objective that encourages models to associate high uncertainty with incorrect token predictions and low uncertainty with correct predictions.  The authors experiment with their fine-tuned models with hallucination detection, selective generation, and out-of-domain detection tasks on CoQA, TriviaQA, and OK-VQA datasets. They show that uncertainty-aware model tuning can provide more reliable uncertainty estimates than standard fine-tuning and improves alignment between uncertainty estimates and response quality.

**Strengths:**

1. The proposed training objective (UA-CLM loss function) shows improved uncertainty estimation in terms of token entropy, perplexity, predictive entropy, and semantic entropy compared to standard fine-tuning.
2. The set of metrics they chose for the experiment is comprehensive, and the gap between the uncertainty-aware and standard fine-tuning is notable.
3. They performed experiments to show the new objective does not negatively impact the generation quality (compared to standard fine-tuning).

**Weaknesses:**

1. This paper lacks a comprehensive literature review, and several claims are unsupported. For instance, the paper does not mention existing works on calibration in long-form language model generations, while many relevant studies exist (e.g., https://arxiv.org/pdf/2310.19208, https://arxiv.org/abs/2404.00474). Additionally, “confabulation” is presented as a subcategory of hallucinations, though it is replacing the hallucination term as it better describes a model’s state of fabricating inaccuracies from a clinical perspective. The authors state they are focusing on confabulation but do not specify which aspects of hallucinations they are addressing, so clarifying the definition and usage of "confabulation" in the context of their work is helpful. Also, they assert that LLMs are often poorly calibrated, though larger models have shown improved calibration compared to smaller ones (https://arxiv.org/abs/2207.05221). Finally, Supervised fine-tuning (SFT) may exacerbate hallucinations not only through overfitting but also due to potential conflicts between the SFT data and the model’s pre-existing knowledge, leading to further inconsistencies. Overall, a more thorough literature review and attention to the accuracy of statements would benefit the paper.
2. The paper only compares the UA fine-tuned models with standard fine-tuned models and does not conduct comparisons with the vanilla, pre-fine-tuning model. So, including comparisons with the pre-fine-tuning model as an additional baseline could be helpful.
3. The experiments do not clarify how token-level labels were assigned for long-form generation tasks. For instance, while TriviaQA is a short-form QA task, models often generate lengthy answers. It is unclear whether target answers in fine-tuning were based on raw dataset responses or if adjustments were made to adapt these answers to LM-style outputs. Please provide more details on their data preparation process, specifically how different responses at different lengths have been handled.
4. The tuned model should be evaluated across additional capabilities, such as reasoning, creative writing, and coding, to ensure these skills are unaffected by the proposed technique; such experiments are currently missing.
5. The BioASK dataset is used for the out-of-domain detection task, but it is unclear if this data is truly out of the models' training knowledge or if it was part of the pre-training data and how this can affect the findings. Providing information on how authors verified that the BioASK dataset was indeed out-of-domain for their models could be helpful.

**Questions:**

Please refer to the weaknesses section for the questions and clarifications requested.

---

> ### Author Response · Authors · 2024-11-20
> **Response to Reviewer RnpC (Part 1)**
>
> Dear Reviewer RnpC,
>
> Thank you for your thoughtful review and valuable suggestions to improve our manuscript. We are glad that you found our proposed training objective shows improved uncertainty estimation, and acknowledging the comprehensive set of metrics chosen for the experiments and notable gap between uncertainty-aware and standard fine-tuning. We have carefully considered your feedback and suggestions and address your concerns below.
>
> > **[W1.1]:** *"This paper lacks a comprehensive literature review, and several claims are unsupported. For instance, the paper does not mention existing works on calibration in long-form language model generations,....."*
>
> Thank you for sharing the references to these recent works on calibration of LLMs. We have included the relevant studies in the related works section including the ones you mentioned in the revised manuscript (updated text highlighted in blue in the second paragraph of Section 2.2).
>
> Furthermore, we have included a study in the **Appendix pages 20-21 (Figures 8-10 and Figures 11-13)** that examines how the proposed loss function calibrates model uncertainty for correctly and incorrectly generated tokens as the loss converges. The findings from this study reveal that our proposed method significantly improves model calibration, supporting the claims of our proposed loss function objectives. With standard CLM, uncertainty decreases for both correct and incorrect tokens, indicating a miscalibration and overconfidence issue. In contrast, our proposed UA-CLM increases uncertainty for incorrect tokens and decreases it for correct tokens, as shown in Figure 8, thereby enhancing the reliability of uncertainty estimates.
>
> > **[W1.2]:** *"Additionally, “confabulation” is presented as a subcategory of hallucinations,....so clarifying the definition and usage of "confabulation" in the context of their work is helpful"*
>
> We adopt the definition of "confabulation" from existing literature, specifically following [[Farquhar et al. 2024](https://www.nature.com/articles/s41586-024-07421-0)] , where they define "confabulations" as arbitrary and incorrect generations. According to [Farquhar et al. 2024], "confabulations" are a subset of hallucinations, characterized by LLMs making fluent claims that are both wrong and arbitrary, meaning the answer is sensitive to irrelevant details such as random seed. This terminology is derived from neuroscience literature [Berrios 1998], which describes it as "false narratives uttered by subjects intent on 'covering up' for a putative memory deficit". Model uncertainty quantification has proven to be a useful indicator for detecting confabulation instances, especially when the model encounters queries outside its knowledge base acquired from the training data distribution. We will include these details and definitions in the revised version of the paper to provide better clarity and context.
>
> > **[W1.3]:** *"Also, they assert that LLMs are often poorly calibrated, though larger models have shown improved calibration compared to smaller ones"*
>
> Our observation is inspired from more recent works [Xiong et al. 2024, Yang et al. 2024, Groot et al. 2024] cited in the introduction section of our manuscript. [Xiong et al. 2024](https://arxiv.org/pdf/2306.13063) show that LLMs including GPT-4 and LLaMA 2 Chat models tend to overconfident, they also find that the model capability scales up, both calibration and failure prediction performance improve similar to the reference you mentioned, but they mention they are yet still far from ideal performance. LLMs such as Llama-2 and Vicuna also exhibit significant overconfidence bias in incorrect answers as evaluated on TruthfulQA dataset [[Yang et al. 2024](https://arxiv.org/pdf/2405.16856)]. VLMs such as GPT-4v and Gemini Pro Vision model also have shown issues of over confidence and poor calibration in their generations [[Groot et al. 2024](https://aclanthology.org/2024.trustnlp-1.13/)].
>
> Ref:
>
> [Farquhar et al. 2024] Farquhar, Sebastian, Jannik Kossen, Lorenz Kuhn, and Yarin Gal. "Detecting hallucinations in large language models using semantic entropy." Nature 630, no. 8017 (2024)
>
> [Berrios 1998] Berrios, German E. "Confabulations: a conceptual history." Journal of the History of the Neurosciences 7.3 (1998)
>
> [Xiong et al. 2024] Xiong, Miao, et al. "Can LLMs Express Their Uncertainty? An Empirical Evaluation of Confidence Elicitation in LLMs." ICLR 2024.
>
> [Yang et al. 2024] Yang, Haoyan, Yixuan Wang, Xingyin Xu, Hanyuan Zhang, and Yirong Bian. "Can We Trust LLMs? Mitigate Overconfidence Bias in LLMs through Knowledge Transfer." arXiv preprint arXiv:2405.16856 (2024)
>
> [Groot et al. 2024] Groot, T. and Valdenegro-Toro, M., 2024, May. Overconfidence is Key: Verbalized Uncertainty Evaluation in Large Language and Vision-Language Models. In Proceedings of TrustNLP Workshop at NAACL 2024.
>
> response continued below...

---

> ### Author Response · Authors · 2024-11-20
> **Response to Reviewer RnpC (Part 2)**
>
> > **[W1.4]:** *"Supervised fine-tuning (SFT) may exacerbate hallucinations not only through overfitting but also due to potential conflicts between the SFT data and the model’s pre-existing knowledge, leading to further inconsistencies"*
>
> We acknowledge that full fine-tuning regime, where all model parameters are updated may exacerbate hallucinations and inconsistencies as reported in a recent study [[Gekhman et al. 2024](https://arxiv.org/pdf/2405.05904)]. However, to mitigate these issues, we employ parameter-efficient fine-tuning using LoRA adapters, where less than 1% of the model parameters are fine-tuned. This approach helps preserve the pre-trained model's knowledge and reduces the risk of overfitting and conflicts, thereby maintaining the integrity and consistency of the model's outputs. We will include this explanation in the revised manuscript to clarify how our method addresses these potential issues.
>
> | Model       |   | Percentage of trainable parameters |
> |-------------|---|------------------------------------|
> | Llama-2-7B  |   | 0.83%                              |
> | Llama-2-13B |   | 0.67%                              |
> | Gemma-2B    |   | 1.03%                              |
> |                       |   |                                         |
>
> >**[W2]:** *"The paper only compares the UA fine-tuned models with standard fine-tuned models and does not conduct comparisons with the vanilla, pre-fine-tuning model. So, including comparisons with the pre-fine-tuning model as an additional baseline could be helpful."*
>
> Thank you for the suggestion. We have included a comparison of our results with the pre-fine-tuning model baseline, as well as with the related fine-tuning methods (Calibration-Tuning (CT) and Unlikelihood training (ULT)) suggested by Reviewer gRZf, in the table below. These comparisons will be included in the revised manuscript.
>
> **Table: Hallucination detection AUROC (comparison of proposed methods with pre-finetuned baseline and other related methods)**
>
> | Model        |   | Fine-tuning Method         |   | AUROC ↑         |            |                    |                  |
> |--------------|---|----------------------------|---|-----------------|------------|--------------------|------------------|
> |              |   |                            |   |  Token Entropy  | Perplexity | Predictive Entropy | Semantic Entropy |
> | Llama-2-7B   |   | Pre-trained baseline |   |      0.5813    |    0.6324   |       0.6686       |      0.7467      |
> |              |   | CLM                                  |   |      0.6252     | 0.6320    |       0.6635        |      0.6889      |
> |              |   | ULT [Welleck et al. 2020]  |   |      0.5790     |   0.5915   |       0.6793       |      0.6495      |
> |              |   | CT [Kapoor et al. 2024]    |   |      0.6175     |   0.6571   |       0.6706       |      0.7292      |
> |              |   | UA-CLM (ours)              |   |    **0.6955**   | **0.7398** |     **0.7413**     |    **0.7741**    |
> |              |   |                            |   |                 |            |                    |                  |
> | Llama-2-13B  |   | Pre-trained baseline                        |   |      0.6027     |   0.6404   |       0.6679       |      0.7111      |
> |              |   | CLM                                  |   |      0.6302     |   0.6348   |       0.6815       |      0.6910      |
> |              |   | ULT [Welleck et al. 2020]  |   |      0.6323     |   0.6523   |       0.6883       |      0.7158      |
> |              |   | CT [Kapoor et al. 2024]    |   |      0.5299     |   0.5599   |       0.6072       |      0.6958      |
> |              |   | UA-CLM (ours)              |   |    **0.6701**   | **0.7255** |     **0.7363**     |    **0.7694**    |
> |              |   |                                       |   |                        |                   |                         |                       |
>
> **Table: Quality of generated text (comparison of proposed methods with pre-finetuned baseline and other related methods)**
> | Finetuning   Method       | Llama-2-7B |          |   | Llama-2-13B |          |
> |---------------------------|------------|----------|---|-------------|----------|
> |                           |   **ROUGE-L ↑**   | **Accuracy ↑**  |   |   **ROUGE-L ↑**    | **Accuracy ↑**  |
> | Pre-trained baseline            |   0.7449   | 0.8350 |   |    0.7832   |  0.8550  |
> | CLM                       |   **0.8886**   |  0.9253  |   |    0.9106   |  0.9406  |
> | ULT [Welleck et al. 2020]    |   0.8409   |  0.8950  |   |    0.8771   |  0.9250  |
> | CT [Kapoor et al. 2024]   |   0.7437   |  0.8125  |   |    0.8022   |  0.8725  |
> | UA-CLM (ours)             |   0.8882   |  **0.9264**  |   |    **0.9118**   |  **0.9461**  |
> |                           |            |          |   |             |          |
>
> response continued below....

---

> ### Author Response · Authors · 2024-11-20
> **Author Response to Reviewer RnpC (Part 3)**
>
> >**[W3]:** *"The experiments do not clarify how token-level labels were assigned for long-form generation tasks. For instance, while TriviaQA is a short-form QA task, models often generate lengthy answers...."*
>
> We would like to clarify that we fine-tune the models for generic causal language modeling (CAUSAL_LM) task in autoregressive manner that predict the next token in a sequence based on the preceding tokens. In the CAUSAL_LM task, labels are created directly from the prompt itself by using the subsequent tokens in the sequence as the target labels for prediction. For the QA task, we process the data sample using the prompt template described in the Appendix A.1.2 (page 15). For each position in the prompt sequence, the model takes the preceding tokens as input and the subsequent token as the label by progressively shifting the window of context during the fine-tuning process.
>
> We set the eos_tokens for different tasks depending on short-form or long-form generation as follow:
>
> | Text generation   task        |   | eos_tokens                                            |   |   |   |   |
> |-------------------------------|---|-------------------------------------------------------|---|---|---|---|
> |                               |   |                                                       |   |   |   |   |
> | phrase-length   generation    |   | ['**.**', '**,**', '**\n**', tokenizer.eos_token_id]  |   |   |   |   |
> | sentence-length   generation  |   | ['**.**', '**\n**', tokenizer.eos_token_id]           |   |   |   |   |
> | paragraph-length   generation |   | ['**\n**', tokenizer.eos_token_id]                    |   |   |   |   |
>
> For evaluation of short-form QA tasks, where we assess the correctness of generated response against the ground-truth sentence, we use the additional prompt prefix: *"Answer the following question as briefly as possible"*.  This prefix, as described in the prompt template in Appendix A.1.2, ensures that the model generates concise answers in addition to cleaning the generated responses by utilizing the eos_tokens described above.
>
> We have also evaluated our method on long-form QA task using ELI5 dataset, please see our results in our response [W1] to Reviewer gRZf.
>
> >**[W4]:** *"The tuned model should be evaluated across additional capabilities, such as reasoning, creative writing, and coding, to ensure these skills are unaffected by the proposed technique"*
>
> We have provide experimental results for biography generation, a paragraph-level writing task in our response below (Part 4) and included in the revised manuscript.  Since, we fine-tune the model for causal language modeling (CAUSAL_LM) task that involves next-token prediction given a set of preceding tokens in a sentence, the learning should be transferable, thereby encouraging generalizability. As such, our focus has been to develop uncertainty-aware fine-tuning in CAUSAL_LM task setup to maximize the applicability to various higher-level tasks for reliable uncertainty quantification. We had chosen one such higher level task (QA) to show the effectiveness of the solution. Our choice of application was made to make the comparison with similar published works on uncertainty estimation in natural language generation [Kuhn et al. 2023, Lin et al. 2024, Fadeeva et al. 2023] straightforward. We believe additional applications can be considered for evaluation as an extension of this research as the proposed method is generalizable. Please see the experimental results demonstrated on biography generation in our response below (Part 4).
>
> >**[W5]:** *"The BioASK dataset is used for the out-of-domain detection task, but it is unclear if this data is truly out of the models' training knowledge... Providing information on how authors verified that the BioASK dataset was indeed out-of-domain for their models could be helpful.**
>
> We selected the BioASK dataset for the out-of-domain detection task after validating that the text generation quality metrics on a pre-trained Llama-2 model were extremely low compared to the other datasets used in our study (CoQA, TriviaQA), as shown in the table below. This validation suggested that the BioASK data was out of the Llama-2 model's training knowledge. However, it is not possible to definitively determine if it was part of the Llama-2 model's training data, as this information is not publicly available. We will include these validation details in the Appendix of the revised manuscript.
>
> | Datasets                              |   | ROUGE-L ↑ | Accuracy ↑ |
> |---------------------------------------|---|-----------|------------|
> | **Llama-2-7B (pre-trained baseline)** |   |           |            |
> | BioASK                                |   | 0.0422    | 0.0530     |
> | CoQA                                  |   | 0.7449    | 0.8350     |
> | TriviaQA                              |   | 0.6654    | 0.7048     |
>
> response continued below....

---

> ### Author Response · Authors · 2024-11-22
> **Author Response to Reviewer RnPC (Part 4)**
>
> > **Update to [W4]:** *"The tuned model should be evaluated across additional capabilities, such as reasoning, creative writing, and coding, to ensure these skills are unaffected by the proposed technique"*
>
> We conducted experiments for biography generation, a long-form paragraph-level generation task, following the recent works from [Band et al. (2024)] and [Liu et al. (2024)]. The Llama-2-7B models fine-tuned with CLM and UA-CLM on CoQA were given prompts to write biographies of popular figures, whose names sourced from BioGen [Min et al. 2023]. The generated responses were compared against those obtained from GPT-4 using the same prompts, which served as the ground truth for evaluation. The results, provided in the table below, show that the generated response quality of both UA-CLM and CLM is similar. Additionally, there is a significant improvement in calibration error and uncertainty quality, as quantified by ECE and hallucination detection AUROC, for UA-CLM fine-tuned model.
>
> | Long-form generation task | Method | BERT score (F1) ↑ |    ECE ↓   |   |    AUROC ↑    |            |                  |
> |--------------------------------|:------:|:---------:|:----------:|:-:|:-------------:|------------|------------------|
> |                                |        |           |            |   | token entropy | perplexity | semantic entropy |
> | **Biography generation**       |        |           |            |   |               |            |                  |
> | Llama-2-7B                     | CLM    |   0.7394  |   0.2511   |   |     0.5653    |   0.5793   |      0.5281      |
> |                                | UA-CLM |   0.7405  | **0.1713** |   |   **0.6135**  | **0.6123** |    **0.5354**    |
> |                                |        |           |            |   |               |            |                  |
>
>  [Band et al. (2024)] Band, Neil, Xuechen Li, Tengyu Ma, and Tatsunori Hashimoto. "Linguistic Calibration of Language Models." ICML 2024.
>
> [Liu et al. (2024)] Liu, Xin, Muhammad Khalifa, and Lu Wang. "LitCab: Lightweight Language Model Calibration over Short-and Long-form Responses." ICLR 2024.
>
> [Min et al. 2023] Min, Sewon, et al. "FActScore: Fine-grained Atomic Evaluation of Factual Precision in Long Form Text Generation." The 2023 Conference on Empirical Methods in Natural Language Processing.

---

> > ### Author Response · Authors · 2024-12-02
> >
> > Dear Reviewer **RnPC**,
> >
> > We have made every effort to address all your concerns and questions in our responses above. We kindly request that you review our rebuttal and let us know if you have any additional questions. If you find that our rebuttal has satisfactorily addressed your concerns, we would be grateful if you could consider updating your evaluation. Your feedback is invaluable to us, and we deeply appreciate the time and effort you have dedicated to reviewing our work. Thank you once again, and we look forward to hearing back from you soon.

---

### Official Review · Reviewer_gRZf · 2024-11-09

**Soundness:** 2
**Presentation:** 2
**Contribution:** 3
**Rating:** 5
**Confidence:** 4

**Summary:**

This paper proposes an uncertainty-aware objective function for causal language model (LM) optimization. The loss encourages the predicted token distribution to have high certainty (low entropy) for correctly predicted tokens and low certainty (high entropy) for incorrectly predicted tokens. Experiments show fine-tuning LMs with the proposed objective leads to improved uncertainty estimates when combined with commonly used uncertainty estimators (mean token entropy, perplexity, predictive entropy, semantic entropy) in several freeform short answer settings.

**Strengths:**

1. The proposed objective is straightforward and simple to adopt in existing frameworks and codebases.
2. Numerous experiments are presented that assess the reliability of LM confidence estimates in several scenarios. This includes hallucination detection (AUROC), selective generation (AUARC), and detecting out-of-domain queries. The ability of the proposed loss to improve uncertainty estimates under several estimators is compelling.
3. Experiments are conducted with LMs and VLMs demonstrating the generality of the proposed loss.

**Weaknesses:**

1. In NLG settings, uncertainty can arise for other reasons than a lack of answer confidence such as semantic equivalence of different paraphrases. Both would appear as incorrect token predictions and are therefore handled identically within the proposed loss. It is plausible that this could lead to significantly degraded performance in longer-form settings due to increasing entropy when only a few options are plausible. Unfortunately, all experiments are in short-form QA settings, leaving this issue unaddressed. This is a significant gap given the motivation to "achieve reliable and well-calibrated uncertainty quantification in open-ended and free-form natural language generation".
2. No alternative fine-tuning baselines are presented. All experiments compare standard fine-tuning to fine-tuning LMs using the proposed loss. This makes it difficult to understand the effectiveness of the proposed approach amidst alternatives from the literature. The post-hoc rescaling works (cited in the paper) appear to be at least one source of reasonable baseline. Other reasonable baselines include simply not computing loss over "incorrect tokens", or applying unlikelihood training to incorrectly predicted tokens, a method that appears closely related to the current work [1]. Alternative lines of closely related work detect hallucinations with probes [2, 3], directly fine-tune LMs to abstain when uncertain [4, 5], or fine-tune LMs to provide calibrated linguistic statements of confidence [6]. In many settings, it may be more useful to have LMs confidently abstain as opposed to outputting high-entropy token distribution, a trade-off that is not discussed in this work.
3. Key details of the proposed loss function are unclear. In particular, the paper does not provide a precise definition of an "incorrect token prediction," a core element of the approach. My assumption is these are tokens where the argmax of the predicted distribution is not equal to the ground truth, but I'm unsure. Other choices should also be more clearly motivated. Is there a particular reason to use tanh over other functions mapping to [0, 1] such as x / (1 + x)? In general, a more detailed analysis of the proposed loss would be beneficial and aid understanding. Examples include visualizations of the ranges of the loss over correctly and incorrectly predicted tokens and analysis of gradients and their analytic form.

### References

1. https://arxiv.org/abs/1908.04319
2. https://arxiv.org/abs/2207.05221
3. https://arxiv.org/abs/2406.15927
4. http://arxiv.org/abs/2403.05612
5. https://aclanthology.org/2024.uncertainlp-1.1
6. https://arxiv.org/abs/2404.00474

**Questions:**

1.  Does the proposed loss degrade performance in longer-form settings?
2.  How does the proposed approach compare to alternative methods like probes and fine-tuning LMs to abstain?
3.  Does removing incorrect generations have similar effects due to not fine-tuning LMs to be confident in these tokens?
4. Is there a motivation for tanh over alternatives?

**Edit: Post Author Response**

I have reviewed the author's response. Many of my questions have been addressed and I have raised my score accordingly.
My primary remaining concern is that the experiments do not demonstrate the approach generalizes beyond short-form QA settings where quality may be significantly impacted by increased entropy.

---

> ### Author Response · Authors · 2024-11-18
> **Response to Reviewer gRZf (Part 1)**
>
> Dear  Reviewer gRZf,
>
> Thank you for your detailed review and constructive feedback. We appreciate your recognition of the strengths of our work for it ease of adoption in existing frameworks, acknowledgment of our numerous compelling experiments demonstrating the generality of the proposed loss with LMs and VLMs. We would like to address your concerns and questions below.
>
> > **[W1]:** *"In NLG settings, uncertainty can arise for other reasons than a lack of answer confidence such as semantic equivalence of different paraphrases."*
>
> We acknowledge that uncertainty in NLG settings can arise due to the semantic equivalence of different phrases. While our proposed loss aims to optimize the predicted token distribution to achieve well-calibrated uncertainties, we have observed that the quality of semantic uncertainty [Kuhn et al. (2023)] also improves significantly with our approach as seen in the results in Table 1 and Table 3; Figure 1 and Figure 2 in the manuscript. This improvement results from the reliable likelihood scores from each sentence and the clustering of sentences with similar meanings. While the focus of this work has been on calibrating token-level uncertainty, we believe it sets the stage for future exploration of calibrating sentence-level uncertainty as mentioned in the third paragraph of discussion section in our paper.
>
> > **[W1]:** *"It is plausible that this could lead to significantly degraded performance in longer-form settings due to increasing entropy when only a few options are plausible. Unfortunately, all experiments are in short-form QA settings, leaving this issue unaddressed."*
>
> We agree that our experiments focus on short-form QA tasks involving sentence-length text generation, similar to the setups and tasks used in the literature [Fadeeva et al. 2023; Kuhn et al. 2023], where entropy-based uncertainty metrics in NLG are studied. For better clarity, we will include a note in Section 4 of the revised manuscript to specify that our evaluations are conducted on sentence-length generation tasks.
>
> Based on your feedback, we evaluated on a long-form QA task using ELI5-Category dataset [Fan et al. (2019)]. The results are provided in table below, we utilized Llama-2-7B model that was fine-tuned with CLM and UA-CLM loss on CoQA dataset. While there is improvement in ECE and AUROC scores with UA-CLM compared to CLM, we believe that more advanced estimators for semantic meaning equivalence of paragraphs will be necessary for accurate paragraph-length uncertainty estimation.
>
> | Dataset / Model      | Method |   | ECE ↓      |   | AUROC ↑            |                  |
> |---------------|--------|---|------------|---|--------------------|------------------|
> |  **ELI5-Category long-form QA**             |        |   |            |   | Predictive Entropy | Semantic Entropy |
> |  Llama-2-7B | CLM    |   | 0.3103     |   | 0.5078             | 0.5393           |
> |               | UA-CLM |   | **0.1833** |   | **0.5172**         | **0.5588**       |
>
> Ref:
>
> [Fan et al. (2019)] Fan, Angela, et al. "ELI5: Long Form Question Answering." Proceedings of the 57th Annual Meeting of the Association for Computational Linguistics. 2019.
>
> [Fadeeva et al. 2023] Fadeeva, E., Vashurin, R., Tsvigun, A., Vazhentsev, A., Petrakov, S., Fedyanin, K., Vasilev, D., Goncharova, E., Panchenko, A., Panov, M. and Baldwin, T., 2023, December. LM-Polygraph: Uncertainty Estimation for Language Models. In Proceedings of the 2023 Conference on Empirical Methods in Natural Language Processing: System Demonstrations (pp. 446-461).
>
> [Kuhn et al. (2023)] Kuhn, Lorenz, Yarin Gal, and Sebastian Farquhar. "Semantic Uncertainty: Linguistic Invariances for Uncertainty Estimation in Natural Language Generation." ICLR 2023.

---

> ### Author Response · Authors · 2024-11-18
> **Response to Reviewer gRZf (Part 2)**
>
> > **[W2]:** *"No alternative fine-tuning baselines are presented....."*
>
> Thank you for your suggestions and related references. We followed your advice and compared our approach with the suggested baselines: UnLikelihood Training (ULT) [Welleck et al. (2020)] and Calibration Tuning (CT) [Kapoor et al. (2024)]. For ULT, we implemented their loss function as described in the paper and fine-tuned using LoRA similar to CLM and UA-CLM; and for CT, we utilized the authors' code from their repository and their fine-tuned models hosted on Hugging Face. The results are presented in Tables below, showing UA-CLM outperforms the compared baselines. We will include these baselines and the comparison results in the revised version of our manuscript.
>
> **Table 1: Hallucination detection AUROC**
>
> | Model        |   | Fine-tuning Method         |   | AUROC ↑         |            |                    |                  |
> |--------------|---|----------------------------|---|-----------------|------------|--------------------|------------------|
> |              |   |                            |   |  Token Entropy  | Perplexity | Predictive Entropy | Semantic Entropy |
> | Llama-2-7B   |   | CLM                        |   |      0.6252     |    0.632   |       0.6635       |      0.6889      |
> |              |   | ULT [Welleck et al. 2020]  |   |      0.5790     |   0.5915   |       0.6793       |      0.6495      |
> |              |   | CT [Kapoor et al. 2024]    |   |      0.6175     |   0.6571   |       0.6706       |      0.7292      |
> |              |   | UA-CLM (ours)              |   |    **0.6955**   | **0.7398** |     **0.7413**     |    **0.7741**    |
> |              |   |                            |   |                 |            |                    |                  |
> | Llama-2-13B  |   | CLM                        |   |      0.6302     |   0.6348   |       0.6815       |      0.6910      |
> |              |   | ULT [Welleck et al. 2020]  |   |      0.6323     |   0.6523   |       0.6883       |      0.7158      |
> |              |   | CT [Kapoor et al. 2024]    |   |      0.5299     |   0.5599   |       0.6072       |      0.6958      |
> |              |   | UA-CLM (ours)              |   |    **0.6701**   | **0.7255** |     **0.7363**     |    **0.7694**    |
>
>
> **Table 2: Quality of generated text**
>
>
> | Finetuning   Method       | Llama-2-7B |          |   | Llama-2-13B |          |
> |---------------------------|------------|----------|---|-------------|----------|
> |                           |   **ROUGE-L ↑**   | **Accuracy ↑**  |   |   **ROUGE-L ↑**    | **Accuracy ↑**  |
> | **CoQA**                  |            |          |   |             |          |
> | CLM                       |   **0.8886**   |  0.9253  |   |    0.9106   |  0.9406  |
> | ULT [Welleck et al. 2020]    |   0.8409   |  0.8950  |   |    0.8771   |  0.9250  |
> | CT [Kapoor et al. 2024]   |   0.7437   |  0.8125  |   |    0.8022   |  0.8725  |
> | UA-CLM (ours)             |   0.8882   |  **0.9264**  |   |    **0.9118**   |  **0.9461**  |
> |                           |            |          |   |             |          |
> | **TriviaQA**              |            |          |   |             |          |
> | CLM                       |   0.6037   |  0.6529  |   |    0.6588   |  0.6967  |
> | ULT [Welleck et al. 2020]    |   0.6121   |  0.6586  |   |    0.6875   |  0.7309  |
> | CT [Kapoor et al. 2024]    |   0.6600   |  0.6987  |   |    0.7018   |  0.7429  |
> | UA-CLM (ours)             |   **0.6679**   |  **0.7108**  |   |    **0.7277**   |  **0.7710**  |
>
>
> >**[W2]:** *"In many settings, it may be more useful to have LMs confidently abstain as opposed to outputting high-entropy token distribution, a trade-off that is not discussed in this work."*
>
> We agree that it can be highly beneficial for LLMs to abstain from answering ambiguous prompts. Our work is motivated by this very principle, aiming to enable models to abstain based on high uncertainty. The thresholds for abstention can indeed vary depending on the specific settings. We have evaluated this ability with **uncertainty-guided selective generation task using AUARC (*Area Under the Accuracy-Rejection Curve*) metric in page 8 of Results section in our paper**. AUARC evaluates the model's decision-making ability of when to generate response and when to abstain based on the model's uncertainty estimates. The results are presented in **Table 1 and Figure 1(b)** of our manuscript.
>
> Ref:
>
> [Welleck et al. (2020)] Welleck, S., Kulikov, I., Roller, S., Dinan, E., Cho, K. and Weston, J., Neural Text Generation With Unlikelihood Training. ICLR 2020.
>
> [Kapoor et al. (2024)] Kapoor, Sanyam, Nate Gruver, Manley Roberts, Arka Pal, Samuel Dooley, Micah Goldblum, and Andrew Wilson. "Calibration-Tuning: Teaching Large Language Models to Know What They Don’t Know." In Proceedings of the 1st Workshop on Uncertainty-Aware NLP (UncertaiNLP 2024).

---

> ### Author Response · Authors · 2024-11-18
> **Response to Reviewer gRZf (Part 3)**
>
> >**[W3]:** *"Key details of the proposed loss function are unclear. In particular, the paper does not provide a precise definition of an 'incorrect token prediction'. My assumption is these are tokens where the argmax...."*
>
> Your understanding is correct that these are tokens where the argmax of the predicted distribution is not equal to the ground truth. We had described the tokens included in $\widetilde{C}$ in lines 211-215 of our submitted paper (lines 216-221 in revised version). We apologize for the lack of clarity on 'incorrect token prediction', we will clarify this in the revised manuscript.
>
> >**[W3]:** *"Is there a particular reason to use tanh over other functions mapping to [0, 1] such as x / (1 + x)?"*
>
> The tanh function was chosen for its smooth gradient properties and bounded output range. However, we acknowledge that other functions mapping to [0, 1], such as x / (1 + x), could also be viable alternatives. We utilized tanh function for this purpose following prior works [Karandikar et al. (2021); Krishnan et al. (2020)] in the uncertainty calibration literature that have shown the utility of tanh for rescaling in the range [0,1] to perform well. We will clarify the motivation behind this design choice in the revised version of our manuscript.
>
> > **[W3]:** *"In general, a more detailed analysis of the proposed loss would be beneficial and aid understanding. Examples include visualizations...."*
>
> Thank you for your thoughtful suggestion. Following your feedback, we have included a detailed analysis of the loss convergence, number of incorrect and correct tokens during the fine-tuning process as loss converges, along with the associated uncertainty estimates in the revised version of our manuscript. Please see the analysis and results in **Appendix Figures 11-13 (page 21) and Figures 8-10 (page 20).**
>
> >**[Q1]:** *"Does the proposed loss degrade performance in longer-form settings?"*
>
> No, calibration and uncertainty quality actually improves. Please see results on ELI5 long-form QA task in response to [W1] above.
>
> >**[Q2]:** *"How does the proposed approach compare to alternative methods like probes and fine-tuning LMs to abstain?"*
>
> We compared our approach with a couple of alternative methods that you suggested: Calibration Tuning (CT) [Kapoor et al. (2024)] and Unlikelihood Training (ULT) [Welleck et al. (2020)]. The results, provided in the tables above in our response to [W2], demonstrate that UA-CLM outperforms these alternative methods.
>
> >**[Q3]:** *"Does removing incorrect generations have similar effects due to not fine-tuning LMs to be confident in these tokens?"*
>
> To address your question, we performed an experiment by fine-tuning with CLM where the loss is computed only on correct token generations. Additionally, we also compared with the Unlikelihood Training (ULT) method that you had suggested, which is related in concept. Both methods have degraded performance compared to our proposed approach in terms of the quality of uncertainty quantification (Hallucination detection AUROC) and ECE. The comparison results are provided in the table below.
>
> | Method                   | ROUGE-L ↑ | Accuracy ↑ | ECE ↓  |   AUROC ↑         |            |                   |                  |
> |--------------------------|---------|----------|---|---------------|-------------|-------------------|------------------|
> |                          |         |          |   | Token Entropy | Perplexity | PredictiveEntropy | Semantic Entropy |
> | **Llama-2-7B (CoQA)**    |         |          |   |               |            |                   |                  |
> | CLM                      | **0.8886**  | 0.9253   | 0.0343  |     0.6252    |   0.6320   |       0.6635      |      0.6889      |
> | CLM (loss computed only with correct token generations) | 0.8858  | 0.9250   | 0.0733  |   0.6672    |   0.6610   |       0.6368      |      0.6601      |
> | ULT                      | 0.8409  | 0.8950   | 0.0588    | 0.5790    |   0.5915   |       0.6793      |      0.6495      |
> | UA-CLM (ours)            | **0.8882**  | **0.9264**   |  **0.0094**   |**0.6955**  | **0.7398** |     **0.7413**    |    **0.7741**    |
>
> >**[Q4]:** *"Is there a motivation for tanh over alternatives?"*
>
> Answered above in our response to [W3].
>
> Thank you again for your valuable suggestions and constructive feedback. Please let us know if you have any further questions or concerns, and any suggestions that will improve our manuscript. If our responses have satisfactorily addressed your concerns and questions, we kindly request that you consider upgrading your evaluation.
>
> Ref:
>
> [Karandikar et al. (2021)] Karandikar, Archit, Nicholas Cain, Dustin Tran, Balaji Lakshminarayanan, Jonathon Shlens, Michael C. Mozer, and Becca Roelofs. "Soft calibration objectives for neural networks.", NeurIPS 2021.
>
> [Krishnan et al. (2020)] Krishnan, R. and Tickoo, O., 2020. Improving model calibration with accuracy versus uncertainty optimization. NeurIPS 2020.

---

> > ### Comment · Reviewer_gRZf · 2024-11-20
> > **Further clarification**
> >
> > Thanks for the additional information, experiments, and clarification. Although I am still reviewing them, I wanted to clarify W1: "It is plausible that this could lead to significantly degraded performance in longer-form settings due to increasing entropy when only a few options are plausible".
> >
> > My primary concern is that while the proposed method is presented as a general approach that "yields better calibrated uncertainty estimates in natural language generation tasks", the experiments do not demonstrate the approach generalizes beyond short-form QA settings, a rather small subset of possible natural language generation tasks. In particular, it seems likely that increasing the entropy of incorrect tokens could cause the LM to output nonsensical (overly random) generations in longer-form settings, thus reducing the overall utility of the approach. Thus, while the additional ELI5 experiments show a correlation between model uncertainty and response quality in longer-form settings, they provide limited optics into the overall degree to which average response quality may (or may not) be degraded when employing the proposed loss.

---

> ### Author Response · Authors · 2024-11-21
> **Response to Reviewer gRZf on clarification to performance in longer-form settings**
>
> Regarding the ELI5 experiments, we apologize for initially misunderstanding your concern about *"degraded performance"* as a degradation in the quality of uncertainty. Consequently, we presented the AUROC and ECE metrics. To address your concern, please find the average response quality, as quantified by the BERT F1 score, in the table below (please let us know if you prefer another alternative metric).
>
> To further clarify, we are fine-tuning the model for the generic CAUSAL_LM task. For instance, with CoQA, the context length of the input query ranges from 400 to 600 tokens, as each data sample includes the story context along with the question (prompt template is described in Appendix A1.2). This effectively makes each data sample a longer paragraph, similar to longer-form settings during fine-tuning process. The number of incorrect and correct tokens during fine-tuning can be observed in **Appendix Figure 8 (page 20)**.
>
> The table below presents the response quality results evaluated on ELI5 long-form QA, demonstrating a 3% improvement in response quality. Additionally, we have shown that our method generalizes well for models fine-tuned with TriviaQA and CoQA in a PEFT setup, with no significant degradation in ELI5 long-form generation.
>
> We have also conducted experiments for biography generation, a long-form paragraph-level generation task, following the recent works from [Band et al. (2024)] and [Liu et al. (2024)]. The Llama-2-7B models fine-tuned with CLM and UA-CLM on CoQA were given prompts to write biographies of popular figures, whose names sourced from BioGen [Min et al. 2023]. The generated responses were compared against those obtained from GPT-4 using the same prompts, which served as the ground truth for evaluation. The results, provided in the table below, show that the generated response quality of both UA-CLM and CLM is similar. Additionally, there is a significant improvement in calibration error and uncertainty quality, as quantified by hallucination detection AUROC, for UA-CLM.
>
> | Evaluation dataset             | Method        |   | BERT (F1) ↑                               |   | BERT (F1) ↑                      |   | BERT (F1) ↑                           |
> |--------------------------------|---------------|---|-------------------------------------------|---|----------------------------------|---|---------------------------------------|
> | **ELI5-Category long-form QA** |               |   | Llama-2-7B  (fine-tuned on ELI5-Category) |   | Llama-2-7B  (fine-tuned on CoQA) |   | Llama-2-7B (fine-tuned on   TriviaQA) |
> |                                | CLM           |   | 0.6922                                    |   | **0.7267**                       |   | 0.7096                                |
> |                                | UA-CLM (ours) |   | **0.7129**                                |   | 0.7130                           |   | **0.7253**                            |
>
>
> | Long-form text generation task | Method | BERT score (F1) ↑ |    ECE ↓   |   |    AUROC ↑    |            |                  |
> |--------------------------------|:------:|:---------:|:----------:|:-:|:-------------:|------------|------------------|
> |                                |        |           |            |   | token entropy | perplexity | semantic entropy |
> | **Biography generation**       |        |           |            |   |               |            |                  |
> | Llama-2-7B                     | CLM    |   0.7394  |   0.2511   |   |     0.5653    |   0.5793   |      0.5281      |
> |                                | UA-CLM |   0.7405  | **0.1713** |   |   **0.6135**  | **0.6123** |    **0.5354**    |
> |                                |        |           |            |   |               |            |                  |
>
> We hope these additional experimental results on ELI5 and biography generation, involving long-form generation address your concerns.

---

> > ### Author Response · Authors · 2024-12-02
> >
> > Dear Reviewer **gRZf**,
> >
> > We have made every effort to address all your concerns and questions in our responses above. We kindly request you to review our rebuttal and let us know if you have any additional questions. If our rebuttal has satisfactorily addressed your concerns, please consider updating your evaluation. Thank you again for taking the time to review our work and we look forward to hear back from you soon.

---

> ### Author Response · Authors · 2024-12-02
> **Response to Reviewer gRZf on generalization beyond short-form QA settings**
>
> > Reviewer **gRZf**: "*My primary remaining concern is that the experiments do not demonstrate the approach generalizes beyond short-form QA settings where quality may be significantly impacted by increased entropy.*"
>
> Thank you for your prompt response and for updating your evaluation. We would like to draw your attention to some key findings that we provided in our responses above addressing your concern. Specifically, we have presented experimental results that evaluate the generalizability of our approach beyond short-form QA settings. These experiments include:
>
> * **Biography Generation Task**: We conducted **paragraph-level text generation** experiments on BioGen.
>
> * **Long-Form QA Task**: We also evaluated our approach on the **ELI5 dataset (paragraph-level generation)**.
>
> The results of these experiments are provided in the Tables in our response above. They demonstrate that uncertainty-aware fine-tuning under PEFT settings with LoRA **generalizes beyond short-form QA tasks without any degradation in the quality of the generated text** (as quantified by the BERT F1 score), while also improving uncertainty estimates quality. We hope these additional experimental results address your concerns. These findings are included in our revised manuscript (last paragraph of Section 4.2 on page 10).
>
> | Long-form text generation task | Method | BERT score (F1) ↑ |    ECE ↓   |   |    AUROC ↑    |            |                  |
> |--------------------------------|:------:|:---------:|:----------:|:-:|:-------------:|------------|------------------|
> |                                |        |           |            |   | token entropy | perplexity | semantic entropy |
> | **Biography generation**       |        |           |            |   |               |            |                  |
> | Llama-2-7B                     | CLM    |   0.7394  |   0.2511   |   |     0.5653    |   0.5793   |      0.5281      |
> |                                | UA-CLM |   0.7405  | **0.1713** |   |   **0.6135**  | **0.6123** |    **0.5354**    |
> |                                |        |           |            |   |               |            |                  |
>
>
> | Long-form QA task             | Method        |   | BERT (F1) ↑                               |   | BERT (F1) ↑                      |   | BERT (F1) ↑                           |
> |--------------------------------|---------------|---|-------------------------------------------|---|----------------------------------|---|---------------------------------------|
> | **ELI5-Category QA** |               |   | Llama-2-7B  (fine-tuned on ELI5-Category) |   | Llama-2-7B  (fine-tuned on CoQA) |   | Llama-2-7B (fine-tuned on   TriviaQA) |
> |                                | CLM           |   | 0.6922                                    |   | **0.7267**                       |   | 0.7096                                |
> |                                | UA-CLM  |   | **0.7129**                                |   | 0.7130                           |   | **0.7253**                            |

---

> > ### Author Response · Authors · 2024-12-03
> >
> > Dear **Reviewer gRZf**,
> >
> > Please let us know if our experiments demonstrating the effectiveness of our approach on **biography generation (a long-form paragraph-level writing task)** and **ELI5-Category (a long-form QA task)**, address your remaining concern about generalization beyond short-form QA settings. We would like to bring these results and findings from our rebuttal to your attention in case they were missed. We are happy to provide any clarifications you may need.

---

### Official Review · Reviewer_Q6st · 2024-11-11

**Soundness:** 3
**Presentation:** 3
**Contribution:** 2
**Rating:** 5
**Confidence:** 3

**Summary:**

This paper aims to equip LLMs with the capability to generate natural language text along with reliable uncertainty estimates. The authors propose an uncertainty-aware language modeling loss focused on a key objective: increasing uncertainty for incorrect token predictions while optimizing for both accuracy and confidence in correct token predictions. They evaluated the uncertainty estimation of LLMs trained with this loss on tasks such as hallucination detection, uncertainty-guided selective generation, and out-of-domain detection. Compared to standard LLMs, the proposed method improves hallucination detection performance while maintaining similar QA accuracy.

**Strengths:**

* This paper is well written and very easy to follow.
* The proposed method, targeting QA and VQA tasks, appears effective for fine-tuning single-task LLMs when a sufficient amount of fine-tuning data is available.
* It is very meaningful to study the model’s uncertainty on its own response.

**Weaknesses:**

* **The experimental setup is very limited**: The experiments were conducted on single-task LLMs with enough fine-tuning data, which may be somewhat restrictive in the current LLM era. I am interested in seeing how effective the proposed approach would be in low-resource settings and in multi-task fine-tuning scenarios.
* **Insufficient critical technical details**: It is unclear what tokens are included in $\widetilde{C}$ and how many there are.
* **Missing latency analysis**: I am interested in seeing the change in latency (training time) introduced by the proposed approach.

**Questions:**

This might be a naïve question, but in Eq2, should $P(w_i|w_{0:i-1})$ be in the correct tokens part, and the $1- P(w_i|w_{0:i-1})$ be in the incorrect part? Apologies if I misunderstood.

---

> ### Author Response · Authors · 2024-11-15
> **Response to Reviewer Q6st (Part 1)**
>
> Dear Reviewer Q6st ,
>
> Thank you for your valuable review and feedback. We appreciate your acknowledgement on the strengths of our work, including the clarity of the paper, the effectiveness of our proposed method on QA and VQA tasks, and the importance of studying model's uncertainty. We address your concerns below:
>
> >**[W1]:** *"I am interested in seeing how effective the proposed approach would be in low-resource settings and in multi-task fine-tuning scenarios"*
>
> We conducted additional experiments to evaluate the generalizability of our method on different QA datasets, simulating low-resource settings. We evaluated the Llama-2-7B model on TriviaQA dataset, which was fine-tuned on different data (small subset of CoQA (20% of the validation set)), and vice-versa. The results for hallucination detection AUROC are presented in the table below. Since we fine-tune the model in a causal language modeling setup that involves next-token prediction, the learning should be transferable, thereby encouraging generalizability. We will include these generalizability experiments in the revised version of the paper.
>
> |                                 | AUROC ↑       |            |                    |                  |
> |---------------------------------|---------------|------------|--------------------|------------------|
> | Method                          | Token Entropy | Perplexity | Predictive Entropy | Semantic Entropy |
> | Llama-2-7B (**CoQA --> TriviaQA**)  |               |            |                    |                  |
> | CLM                             |     0.7201    |   0.7847   |       0.7532       |      0.7874      |
> | UA-CLM (ours)                          |   **0.8271**  | **0.8261** |      **0.788**     |    **0.8146**    |
> |                                 |               |            |                    |                  |
> | Llama-2-7B (**TriviaQA --> CoQA**)  |               |            |                    |                  |
> | CLM                             |     0.5952    |   0.6349   |       0.6665       |      0.7146      |
> | UA-CLM (ours)                         |   **0.6456**  | **0.6824** |     **0.7154**     |    **0.7528**    |
>
> **Update:** We also conducted experiments for biography generation (a long-form generation task) using the Llama-2-7B model fine-tuned with CLM and UA-CLM on CoQA. We gave the prompts to write biographies of popular figures, whose names are sourced from BioGen [Min et al. 2023]. The generated responses were compared against those obtained from GPT-4 using the same prompts for evaluation. The results in the table below show significant improvement in calibration error and uncertainty quality, as quantified by ECE and hallucination detection AUROC, demonstrating our approach is effective in multi-task fine-tuning scenarios.
>
> | Long-form generation task | Method | BERT score (F1) ↑ |    ECE ↓   |   |    AUROC ↑    |            |                  |
> |--------------------------------|:------:|:---------:|:----------:|:-:|:-------------:|------------|------------------|
> |                                |        |           |            |   | token entropy | perplexity | semantic entropy |
> | **Biography generation**       |        |           |            |   |               |            |                  |
> | Llama-2-7B (**CoQA --> BioGen**)                    | CLM    |   0.7394  |   0.2511   |   |     0.5653    |   0.5793   |      0.5281      |
> |                                | UA-CLM |   0.7405  | **0.1713** |   |   **0.6135**  | **0.6123** |    **0.5354**    |
> |                                |        |           |            |   |               |            |                  |
>
> >**[W2]:** *"It is unclear what tokens are included in $\widetilde{C}$ and how many there are."*
>
> We understand the need for more detailed information regarding the tokens included in our fine-tuning process. We had described the tokens included in $\widetilde{C}$ in page 5 (lines 216-221) in the revised manuscript. $\widetilde{C}$ includes the tokens that are incorrectly predicted by the model, as compared to the ground-truth tokens in the sequence, in every mini-batch during fine-tuning. Further, based on your feedback we have included a detailed analysis on the number of incorrect tokens and correct tokens during the fine-tuning process, along with the associated uncertainty estimates in **Appendix Figures 8-10 (page 20)** of the revised version of the paper.
>
> Our analysis indicates that both CLM and UA-CLM shows an increase in correct tokens and a decrease in incorrect tokens as fine-tuning progresses, **as illustrated in Figure 8**. With CLM, uncertainty decreases for both correct and incorrect tokens, which suggests miscalibration overconfidence issue. In contrast, our proposed UA-CLM increases uncertainty for incorrect tokens and decreases it for correct tokens **as shown in Figure 9**, thereby improving the reliability of uncertainty estimates.
>
> continued response below...

---

> ### Author Response · Authors · 2024-11-15
> **Response to Reviewer Q6st (Part 2)**
>
> > **[W3]:** *"I am interested in seeing the change in latency (training time) introduced by the proposed approach."*
>
> The training latency and throughput comparison is presented in table below (parameter-efficient fine-tuning with LoRA).  we see a 6-7% increase in training time with our method while the model benefits from improved uncertainty calibration. The latency measurements for both methods were made on similar setup for a fair comparison (Batch size of 2 with an average 512 tokens in a sequence, 4-bit normalized float (nf4) quantization to the model’s parameters and FP16 precision for compute on Intel(R) Xeon Gold 6448H CPU and Nvidia A100 80G GPU, with LoRA fine-tuning (<1% params updated), similar to hyperparameter settings description in Appendix A.1.3). These numbers may vary depending on different setups, but the relative difference should ideally remain consistent.
>
> | Method    |   Latency ↓   | Throughput ↑ | Relative Latency |
> |-----------------|:-------------:|:------------:|:------------------:|
> | **Llama-2-7B**  |               |              |                    |
> | CLM             | 2.63 ms/token | 379 token/s  |          1x          |
> | UA-CLM (ours)          | 2.78 ms/token | 359 token/s  | 1.06x          |
> |                 |               |              |                    |
> | **Llama-2-13B** |               |              |                    |
> | CLM             | 4.50 ms/token | 222 token/s  |        1x            |
> | UA-CLM (ours)          | 4.85 ms/token | 206 token/s  | 1.07x          |
>
>
> >**[Q1]** *"This might be a naïve question, but in Eq2, should $P_{\theta}(w_i | w_{0: i\hbox{-}1})$ be in the correct tokens part, and the $\left(1-P_{\theta}(w_i | w_{0: i\hbox{-}1})\right)$ be in the incorrect part? "*
>
> Thank you for your question. The probability components in Eq. 2 are correctly placed as they are. The desired and ideal outcome in causal language modeling is to achieve a state where every correctly generated token is assigned low predictive uncertainty and high predictive probability i.e., $P_{\theta}(w_i | w_{0: i\hbox{-}1})$=1, reflecting the model’s high confidence in its accuracy. Conversely, for every token that is generated incorrectly, the model should assign high uncertainty and low predictive probability i.e., $P_{\theta}(w_i | w_{0: i\hbox{-}1})$=0, denoting low confidence in these instances. This ensures that the model’s confidence levels and uncertainty estimates are perfectly calibrated with the actual correctness of its predictions. The intuition is that as {$P_{\theta}(w_i | w_{0: i\hbox{-}1})$ $\to$ 0} for all incorrect tokens, the loss decreases {L $\to$ 0}, and similarly, as {$P_{\theta}(w_i | w_{0: i\hbox{-}1})$ $\to$ 1} for all correct tokens, {$\left(1-P_{\theta}(w_i | w_{0: i\hbox{-}1})\right)$ $\to$ 0}, resulting in a loss that approaches 0. We hope this clarifies your question.
>
> If the details from our additional experiments and analysis address your concerns, we kindly request that you consider upgrading your evaluation score. Thank you once again for your time and consideration.

---

> > ### Author Response · Authors · 2024-12-02
> >
> > Dear Reviewer **Q6st**,
> >
> > We have made every effort to address all your concerns and questions in our responses above. We kindly request you to review our rebuttal and let us know if you have any additional questions. If our rebuttal has satisfactorily addressed your concerns, please consider updating your evaluation. Thank you again for taking the time to review our work and we look forward to hear back from you soon.

---

### Author Response · Authors · 2024-11-22
**General response**

We thank all five reviewers for their thoughtful reviews, questions and valuable suggestions. We are glad that the reviewers recognized our proposed uncertainty-aware fine-tuning method to be theoretically sound and interesting idea ( ***Br54***, ***wXAr***), effective and compelling in improving uncertainty estimation (***Q6st***, ***RnpC***, ***gRZf***, ***Br54***, ***wXAr***), and simple to adopt in existing frameworks (***gRZf***). We appreciate the reviewers' recognition of our paper as well-written and easy to follow (***Q6st***, ***wXAr***), with a thorough and nicely structured literature review (***wXAr***). We also appreciate that reviewers found our empirical experiments and results to be comprehensive and rigorous, demonstrating the effectiveness of proposed method (***RnpC***, ***Br54***, ***wXAr***, ***gRZf***).

**We have addressed the comments and questions in individual responses to each reviewer**. We kindly request the reviewers to read our responses to their comments and questions, ask any follow-up questions or provide suggestions. Below is a summary of the additional experiments and updates we have made based on the suggestions from the reviewers:

*	We conducted additional experiments to evaluate the generalizability of our method, demonstrating that it performs well in low-resource settings where fine-tuning data is not available.  **[*Q6st*]**

*	We performed a detailed analysis of the number of correctly and incorrectly generated tokens and their associated uncertainty estimates as the loss converges. This supports our claims and demonstrates that the proposed method improves the reliability of uncertainty estimates. The results are provided in **Appendix Figures 8-13 (pages 20-21)**.  **[*Q6st*, *gRZf*]**

*	We provided latency and throughput measurements, showcasing that there is no significant difference in fine-tuning duration between the proposed method and standard fine-tuning. **[*Q6st*, *Br54*]**

*	We compared our method with a pre-trained baseline and related methods: UnLikelihood Training (ULT) [Welleck et al. (2020)] and Calibration Tuning (CT) [Kapoor et al. (2024)], and demonstrated that our method outperforms these baselines. **[*gRZf*, *RnpC*]**

* We evaluated our method on a long-form QA and biography generation task, to show that our method improves the reliability of uncertainty estimates and the quality of generated responses in long-form generation settings as well. **[*gRZf*, *RnpC*]**

*	We computed the sentence-level expected calibration error (ECE) and demonstrated that our method enables the model to achieve a low ECE. An analysis of the accuracy versus ECE trade-off is provided in **Appendix Figures 14-15 (Page 22)**. **[*wXAr*, *Br54*]**

*	We performed experiments to compute the improvement in another related uncertainty metric, 'eccentricity' [Lin et al. 2023], using our fine-tuning method, for comparison with entropy-based metrics. **[*wXAr*]**

*	Regarding changes in the revised manuscript, we updated the related works section with additional recent works suggested by the Reviewer **[*RnpC*]**. We have consolidated and included additional experimental results based on reviewers feedback listed above in the manuscript. All the text changes are highlighted in blue.

---

### Note · Authors · 2025-01-22

I have read and agree with the venue's withdrawal policy on behalf of myself and my co-authors.